

# On the formation of biogenic secondary organic aerosol in chemical transport models: an evaluation of the WRF-CHIMERE (v2020r2) model with a focus over the Finnish boreal forest

Giancarlo Ciarelli[1], Sara Tahvonen[1], Arineh Cholakian[2], Bruno Vitali[3], Tuukka Petäjä[1] and Federico Bianchi[1]

[1]Institute for Atmospheric and Earth System Research/Physics, Faculty of Science, University of Helsinki, Helsinki, 00014 Finland.
[2]LMD UMR CNRS 8539, ENS, École Polytechnique, Institut Pierre Simon Laplace (IPSL), Route de Saclay, 91128 Palaiseau, France.
[3] Department of Civil, Environmental and Mechanical Engineering, University of Trento, 38123 Trento (TN), Italy.

*Correspondence to*: Giancarlo Ciarelli (giancarlo.ciarelli@helsinki.fi)

**Abstract.** We present an evaluation of the regional chemical transport model (CTM) WRF-CHIMERE (v2020r2) for the formation of biogenic secondary organic aerosol (BSOA) with a focus over the Finnish boreal forest. Formation processes of biogenic aerosols are still affected by different sources of uncertainties, and model's predictions largely varies depending on the levels of details of the adopted chemical and emissions schemes. In this study, air quality simulations were conducted for the astronomical summer of the year 2019 using different organic aerosol (OA) schemes (as currently available in literature) to treat the formation of BSOA. First, we performed a set of simulations in the framework of the volatility basis set (VBS) scheme carrying different assumptions for the treatment of the aging processes of BSOA. The model results were compared against high-resolution (i.e., 1-hour) organic aerosol mass and size distribution measurements performed at the Station for Measuring Ecosystem–Atmosphere Relations (SMEAR-II) site located in Hyytiälä, in addition to other gas-phases species such as ozone ($O_3$), nitrogen oxides ($NO_x$) and BVOCs measurements of isoprene ($C_5H_{10}$) and monoterpenes. We show that WRF-CHIMERE could well reproduce the diurnal variation of the measured OA concentrations for all the investigated scenarios (along with standard meteorological parameters) as well as the increase in concentrations during specific heat waves episodes. However, the modeled OA concentrations largely varied between the schemes use to describe the aging processes of BSOA. Additionally, comparisons with isoprene and monoterpenes air concentrations revealed that the model captured the observed monoterpenes concentrations, but isoprene was largely overestimated, a feature that was mainly attributed to the overstated biogenic emissions of isoprene. We investigated the potential consequences of such an overestimation by inhibiting isoprene emissions from the modeling system. Results indicated that the modeled BSOA concentrations generally increased compared to the base-case simulation with enabled isoprene emissions. We attributed the latest to a shift in the reactions of monoterpenes compounds against available radicals, as further suggested by the reduction in α-pinene modeled air concentrations. Finally, we briefly analyze the differences in the modeled Cloud Liquid Water Content (clwc) among the simulations carrying different chemical scheme for the treatment of the aging processes of BSOA. Model's results indicated



an increase in clwc values at the SMEAR-II site, for simulation with higher biogenic organic aerosol loads, likely as a results of the increased numbered of biogenic aerosol particles capable of activating cloud droplets.

## 1 Introduction

Aerosol particles arising from the terrestrial ecosystem, referred to as biogenic aerosols, often constitute a major fraction of the observed total particulate mass (PM) (Ciarelli et al., 2016; Jiang et al., 2019b). Their contribution to PM can largely vary depending on the specific land use as well as synoptic and local meteorological conditions throughout the year (Guenther et al., 2006, 2012; Oderbolz et al., 2013).

A sub-set of particles of biogenic origin that are directly emitted into the atmosphere, e.g. pollens, mineral dust and sea salt, are usually referred to as primary particles. They show a rather coarse particle diameters and are efficiently removed from the atmosphere via scavenging processes (Jacobson, 2005). The second group of biogenic aerosols, referred to as secondary particles, are produced in the atmosphere as a result of a series of complex chemical reactions on a time scale ranging from seconds to days (Seinfeld and Pandis, 2012).

The main precursor for the secondary organic aerosol mass originates from earth's vegetation that emits several Volatile Organic Compounds (VOCs, Guenther et al., 2012) that are usually in the gas-phase form at the most relevant ambient conditions and emissions highly depends on the local meteorological parameters (e.g. temperature, radiation, soil moisture (Guenther et al., 2012; Peñuelas et al., 2014) as well as on the specific plant ecotype. Modelling studies have shown that isoprene ($C_5H_8$) and monoterpenes are the most abundant organic compound emitted from earth's vegetation (Sindelarova et al., 2014), but estimates highly depend on the model's driving variables and, in particular, on the plant functional types (PFTs) data and emissions factors (EFs) associated with them (Bergström et al., 2014; Jiang et al., 2019a). Once released into the atmosphere VOCs can quickly (seconds to hours) react towards the hydroxyl radical ($^{\cdot}OH$), ozone ($O_3$) and the nitrate radical ($^{\cdot}NO_3$) to produce organic gases with a sufficient low volatility to transition into the particle phase (Xu et al., 2022). The resulting additional particulate mass is widely referred as to secondary organic aerosol (SOA), and, if produced from biogenic volatile organic compounds (BVOCs) such as isoprene and monoterpenes, to biogenic secondary organic aerosol (BSOA). Numerous regional modeling studies, have focused on the formation and characterization of the BSOA component (Aksoyoglu et al., 2011; Bergström et al., 2012; Boy et al., 2022; Cholakian et al., 2022, 2018; Ciarelli et al., 2016; Hodzic et al., 2009, 2009; Zhang et al., 2013). However, such a highly complex system governing the formation of BSOA remains not fully understood.

Initial attempts to implement the formation of secondary organic aerosol in three dimensional chemical transport models (CTMs) made used of chamber SOA yields fitted with two condensable gases to mimic the oxidation products of the parent precursors, so called Odum scheme (Odum et al., 1996), as well a partition coefficient for each for the condensable gases. The Clausius-Clapeyron equation is used to adjust the saturation vapor pressure ($C^*$) based on temperature and using a prescribed set of vaporization enthalpies ($\Delta H_{vap}$). This approach has the advantage of being computationally efficient and suitable to





simulate the total organic mass in a wide range of large scale (regional to global) application but it is limited by the number of the adopted surrogate species (i.e. two) and their prescribed mean molecular weight specifications. To improve the level of details of the organic fraction in chemical transport model, a so-called Volatility Basis Set (VBS) was developed. In the VBS model, the oxidation products of a parent hydrocarbon are distributed across a wide range of volatilities, each of them with a molecular structures derived from the group-contribution approach (Donahue et al., 2011; Donahue et al., 2012). In this

framework, the organic mass is binned in volatility classes, ranging from Extremely Low Volatility Organic Compounds (referred to as ELVOCs), with $C^* < 3 \times 10^{-5}$ µg m$^{-3}$ to VOCs with $C^* > 3 \times 10^{6}$ µg m$^{-3}$ (Bianchi et al., 2019). Such an approach allows tracking the volatility distribution of ambient organic aerosols as well as the degree of oxygenation of the air mass (i.e. oxygen to carbon ration). Additionally, the VBS scheme allows for further chemical reaction of primary, and secondary produced, semi-volatile organic carbon (SVOC) gases (i.e. so called chemical aging) available in the $0.3 < C^* < 300$ µg m$^{-3}$

saturation concentration range. Such computational approach has been successfully applied to corroborate multiple chamber aerosol chemical aging studies that revealed an further increase in SOA concentrations when first-generation oxidation products were further reacted against the ˙OH radical (Donahue et al., 2012).

On a global scale, Tsigaridis et al., 2014 compared model simulation of thirty-one chemical transport models of OA and general circulation models (GCMs) in the framework of the AeroCom phase II. Their results indicated that model simulation of OA

greatly varies between models, mainly due to the increasing complexity of the SOA parameterization and addition of new OA source in recent years. In Europe, a growing number of chemical transport modeling studies have been performed with a focus on the BSOA fraction of OA. Bergström et al., 2012 tested the aging of BSOA in the EMEP model using aging reaction rates constant as proposed by Lane et al., 2008 (i.e. $4.0 \times 10^{-12}$ cm$^3$ molecule$^{-1}$ s$^{-1}$). Model results were evaluated against measurements data available during 2002 and 2007 mainly using filter measurements of organic carbon (OC) as available at

different EMEP rural background sites (at daily and weekly time resolution). Their results suggested that, compared to other aging schemes, accounting for aging reactions of BSOA (PAA method in Bergström et al., 2012) improved model prediction of OC during summertime and at the majority of the sites. Similarly, Zhang et al., 2013 deployed the CHIMERE model with two nested domains covering Europe and Northern France with a 3 km grid resolution for the latest. Aging of BSOA where identical to the aging of the anthropogenic secondary organic aerosol (ASOA) i.e. $1.0 \times 10^{-11}$ cm$^3$ molecule$^{-1}$ s$^{-1}$ (Murphy and

Pandis, 2009), and biogenic emissions were driven with the Model of Emissions of Gases and Aerosols from Nature (MEGAN) model (Guenther et al., 2006). Model simulation of OA were performed for the MEGAPOLI summer campaign of July 2009 and compared against 1-hour aerosol mass spectrometer (AMS) data available in the Greater Paris area (1 urban and 2 suburban sites). Their results indicated that accounting for aging of both anthropogenic and biogenic SVOCs helped to improve the agreement between modeled and observed OA, particularly in terms of the temporal variabilities and occurring times of major

pollutions peaks. Long-range related air masses, however, were overestimated in the model, possibly because of the too aggressive aging chemical scheme in the model. These results were additionally confirmed by a later application of the CHIMERE model during the ChArMEx 2013 campaign conducted in the western Mediterranean basin, i.e. Ersa, Cap Corse (Corsica, France) (Cholakian et al., 2017). Comparison of modeled and observed OA concentrations indicated a large





overestimated of the OA fraction in the model when aging of BSOA where implemented following the same pathway as ASOA

(i.e. $K_{OH} = 1.0 \times 10^{-11}$ cm$^3$ molecule$^{-1}$ s$^{-1}$), suggesting a too aggressive production of low-volatile gases available to rapidly transition in the particle phase. Therefore the author also tested a fragmentation schemes in their VBS model where the oxidization products of the parent hydrocarbon were allowed to fall in an higher saturation vapor pressure range (compared to the parent precursor, i.e. fragmentation), and using a branching ratio for the distribution of the products, i.e. 75% fragmentation and 25% functionalization (Shrivastava et al., 2015). Model results indicated a large reduction in the model positive bias

compared to simulation with non-fragmentation processes and aging of BSOA.

In this study, we focus on the formation and aging processes of BSOA as an important fraction of the total OA in areas that are affected directly, and largely, by biogenic emissions, i.e. the Finnish boreal forest. As new high temporal resolution measurements of OA and biogenic gas-phase compounds are now available, we evaluate 1) the effect of BSOA chemical aging in the WRF-CHIMERE model, 2) the model performance with respect to BVOC emissions, meteorological parameter, photochemistry (i.e. NO$_x$ and O$_3$) and OA mass at the Station for Measuring Ecosystem–Atmosphere Relations (SMEAR-II

(Hari and Kulmala, 2005) site, 3), the sensitivity of OA formation in the model with respect to isoprene emissions, and in particular on BSOA, and 4), the changes in the modeled Cloud Liquid Water Content (clwc) when the treatment of the aging processes of BSOA is accounted for.





## 2 Method

### 2.1 The WRF-CHIMEREv2020r2 Model

The WRF-CHIMEREv2020r2 model, WRF-CHIMERE thereafter (Menut et al., 2021), is a three dimensional CTM capable to simulate physical and chemical processes taking place into the atmosphere, from the injection of emissions in the planetary boundary layer (PBL), to chemical reaction of hundreds of chemical compounds to dry and wet deposition processes. The model has participated in numerous intercomparison exercises (Bessagnet et al., 2016; Ciarelli et al., 2019; Solazzo et al., 2017; Theobald et al., 2019) and it is an active member of the Copernicus Atmosphere Monitoring Service (CAMS) operational ensemble. Recently, it has been upgraded to run "online" with the Weather Research and Forecast (WRFv3.71) model to include the exchange of the aerosol size distribution, among other parameters, between CHIMERE and the meteorological model, i.e. WRF, (Briant et al., 2017; Tuccella et al., 2019). It can be applied at various horizontal resolutions, and it is therefore suitable for both global (hundreds of kilometers) and urban (1 km) scale applications (Bessagnet et al., 2017; Mailler et al., 2017).

Simulations were performed for the astronomical summer of 2019 (i.e., from 15 June 2019 until 31 August 2019) using two domains on a Lambert conformal projection: a first domain covering whole Europe at about 30 x 30 km resolution and a second nested domain centered over Finland at about 10 x 10 km resolution (Figure 1). The chemical mechanism used for the gas-phase chemistry was the MELCHIOR2 scheme (Derognat, 2003), including up to about 120 reactions with updated reaction rates (last updated in 2015). The ISORROPIA thermodynamic model was used to calculate the partitioning of the inorganic aerosol constituents (Nenes et al., 1998) and a logarithmic sectional distribution approach was deployed to treat the size distribution of aerosol particles using 15 bins ranging from 10 nm to 40 µm. The model additionally account for coagulation process (Debry et al., 2007) as well as binary nucleation of sulfuric acid ($H_2SO_4$) and water (Kulmala et al., 1998). The treatment of OA in the model, and specifically of BSOA in the framework of the VBS scheme, is described in details in the next section.

### 2.2 BSOA schemes

The VBS scheme was first implemented in the CHIMERE model for the Mexico City metropolitan area during the MILAGRO 2006 field experiment (Hodzic and Jimenez, 2011) and first applied over Europe for the Metropolitan area of Paris (Zhang et al., 2013). Oxidization products of BVOCs are distributed into four classes of volatility at a saturation concentrations of 1, 10, 100, and 1000 µg m$^{-3}$ (at 300 K) with different mass yields for low-NO$_x$ and high-NO$_x$ conditions based on the work of Hodzic and Jimenez, 2011 (Figure 2), and allocated in a dedicated set to uniquely track their contribution to OA. The resulting gas-phase material, can be further oxidized in the model by the ·OH radical (blue curved arrows in Figure 2) resulting in an increase of 7.5 % in the organic mass to mimic the addition of an oxygen (Robinson et al., 2007) and in a simultaneous shift in volatility by one order of magnitude. In this VBS schemes, also referred to as 1D VBS schemes, a fixed molecular structure, and therefore molecular weights, is assigned to each of the four volatility classes, i.e. 180 g mol$^{-1}$. The enthalpy of evaporation ($\Delta H_{vap}$) of





each BSOA volatility class is also unique and set to 36 kJ mol⁻¹. The mass yield of biogenic aerosol for each of the volatility bins are taken from Hodzic and Jimenez, 2011 (and reference therein) with similar reaction rates for the first oxidation step.

We tested the oxidation of gas-phase biogenic organic material in the SVOC range, i.e., chemical aging, (blue curved arrows in Figure 2), using different reactions as available from current literature. Additionally, we also tested the influence of isoprene emissions on BSOA formation (Table 1). In total, we performed four simulations, as described below:

-    Aging-On-Case-1: Gas-phase organic material of biogenic origin in the SVOC range can react with the ˙OH radical with a reaction rate of 1 x 10⁻¹¹ molecule⁻¹ cm³ s⁻¹ (Murphy and Pandis, 2009; Zhang et al., 2013).

-    Aging-On-Case-2: Gas-phase organic material of biogenic origin in the SVOC range can react with the ˙OH radical with a reaction rate of 4 x 10⁻¹² molecule⁻¹ cm³ s⁻¹ (Bergström et al., 2012; Lane et al., 2008). This simulation also represents the base-case simulation for the evaluation of both meteorological parameters and the photochemistry.

-    Aging-Off: Gas-phase organic material of biogenic origin in the SVOC range does not further react with the ˙OH radical. SVOC species are included in the partitioning equations and/or removed from the system via wet and/or dry deposition.

-    C₅H₈-emissions-Off: Emissions of C₅H₈ are inhibited in the emissions model (i.e. MEGAN). Gas-phase organic material of biogenic origin in the SVOC range can react with the ˙OH radical with a reaction rate of 4 x 10⁻¹² molecule⁻¹ cm³ s⁻¹ (Bergström et al., 2012; Lane et al., 2008), i.e. based on Aging-On-Case-2.

**2.3 Input data**

Annual anthropogenic emissions of black carbon (BC), organic carbon (OC), carbon monoxide (CO), ammonia (NH₃), non-methane volatile organic compounds (NMVOCs), nitrogen oxides (NOₓ) and sulfur dioxide (SO₂) were retrieved from CAMS for the whole year 2019 at 0.1 x 0.1 degree resolution and hourly distributed over the investigated periods (astronomical summer of 2019). Biogenic emissions of NO, isoprene, limonene, α-pinene, β-pinene, ocimene, and humulene (representing the lumped class of sesquiterpenes) were prepared using the MEGAN model version 2.1 (Guenther et al., 2012). Emission rates of 15 plant functional types (PFTs), at an original horizontal resolution of 0.008º × 0.008º, were re-gridded to match the resolution of both the coarse and high-resolution nested domains (i.e. 30 km and 10 km, respectively). Standard emissions rate are adjusted based on several environmental factors, based on local radiation and temperature values (among others variables such as leaf area index (LAI), Guenther et al., 2006).

Meteorological input were simulated with the WRF regional model (v3.71) (Skamarock et al., 2008) forced by National Centers for Environmental Prediction (NCEP) Climate Forecast System Version 2 (http://www.ncep.noaa.gov, last access: 20 February 2023) with a temporal resolution of 6 h, an horizontal resolution of 1 degree, and with the course domain nudged towards the reanalysis data (every 6 hours, i.e. surface grid nudging). Simulation were performed using the Rapid Radiative Transfer Model (RRTMG) radiation scheme (Mlawer et al., 1997), the Thompson aerosol-aware MP scheme to treat the microphysics (Hong et al., 2004) the Monin–Obukhov surface layer scheme (Janjic, 2003), and the NOAA Land Surface Model scheme for land surface physics (Chen and Dudhia, 2001).





Initial and boundary conditions of aerosols and gas-phase constituents were retrieved from the climatological simulations of

180    LMDz-INCA3 (Hauglustaine et al., 2014) and the Goddard Chemistry Aerosol Radiation and Transport (GOCART) model (Chin et al., 2002). For aerosol species, in particular, the model includes inorganic species such as fine and coarse nitrate, ammonium, sulfate, dust, as well as OC and BC.





## 3 Observational data and model evaluation methods

Observational data were taken from at the SMEAR-II station located within the boreal forest of Finland (black cross in Figure
185  1). Common meteorological parameters with a temporal resolution of 1-hour were used to evaluate the performance of the
WRF model, i.e., surface temperature, wind speed, wind direction, relative humidity, and precipitation.

The CHIMERE model was evaluated with gas-phase measurements of isoprene, monoterpenes, ozone and nitrogen oxides (in
dry air) taken at 4.2 meters on a 1 hour temporal resolution. Nitrogen oxides measurements were performed with a
chemiluminescence analyzer (TEI 42 CTL) whereas ozone measurements were performed with the ultraviolet light absorption
190  analyzer (TEI 49 C). Measurements of isoprene and monoterpenes BVOCs were performed with the proton transfer reaction-
mass spectrometry (PTR-MS, Rantala et al., 2015).

The modeled total organic aerosol mass (OA) was compared against Aerosol Chemical Speciation Monitor (ACSM)
measurements available during the astronomical summer of 2019 (i.e. from 15 June 2019 until 31 August 2019). The ACSM
measurers the non-refractory (NR) sub-micrometer particulate matter mass (i.e. material evaporating at 600º) with an
195  aerodynamic diameter less than 1 µm ($PM_1$). A thorough description of the ACSM measurements are available in Heikkinen
et al., 2021. Finally, the modeled particle size distribution was compared against Differential Mobility Particle Sizer (DMPS)
measurements.

Additionally, measurements of total particulate matter with an aerodynamic diameter less than 2.5 µm (i.e., $PM_{2.5}$) were taken
from the Air Quality e-Reporting (AQ e-Reporting) database (Table S1). Daily measurements were extracted for rural stations
200  below 200 meters above sea level (avoiding urban hot spots and complex terrain) located in the most southern regions of the
domain, which are likely to be exposed to higher oxidant levels. Observations at these measurement sites were compared
against the total modeled $PM_{2.5}$ mass available from the course grid (at 30 km). The statistical metrics used for the
meteorological and chemical performance evaluation are reported in Table 2.



## 4 Results

### 4.0 Synoptic context

The summer of 2019 was the second warmest on global scale and was among the 5 warmest measured in Europe since the 16[th] century, producing a regional temperature anomaly close to 2 K (compared to 1981-2010), comparable to that of 2003 (Sousa et al., 2019). Two distinct severe heat wave events occurred during the period considered in this study, the first in late June and the second in late July. Both events were characterized by the presence of a low-pressure system in the North-eastern Atlantic and a ridge extending over Europe, causing persistent anticyclonic conditions, low cloud cover and warm sub-tropical air advection from northern Africa, a configuration typically associated to extreme temperatures (Tomczyk et al., 2017). The synoptic pattern of the late June heat wave was better defined, and affected mainly southwestern Europe, while the late July heat wave could reach further northward toward Scandinavia, affecting also Finland, where a record temperature of 33.2 C was measured on the 28[th] (Villiers, 2020). Besides the dynamical influence, the first event was enhanced by vertical descent of potentially warmer air. Differently, the late July heat wave was driven by diabatic fluxes and surface-atmosphere coupling, a process amplified by the soil moisture deficit produced by the first extreme event (Sousa et al., 2019). In Figure 3 we show the large-scale configuration of the late July event, which strongly affected Finland. On the 19[th] the high pressure was already located on the Iberian Peninsula, and started to expand northwards. On the 25[th] the strongest pressure and temperature anomalies were registered in France and Spain, and the ridge started to influence northern Europe. After the 26[th] the Atlantic low moved East across Great Britain bringing cooler air to continental Europe, which was however still affecting Scandinavia. On the 29[th] Finland started to be influenced by western cold continental air.

### 4.1 Analysis and evaluation of meteorological parameters

Meteorological conditions are a fundamental ingredient to understand the formation, transportation and removal of pollutants (Bianchi et al., 2021; Seinfeld and Pandis, 2012). However, there are not always simultaneously analyzed in CTM applications, and often uncertainties are presented relatively to the underlying gridded emissions and/or chemical mechanics. It is therefore important to characterize also the meteorological conditions and evaluate the key meteorological parameters that are driving the physical and chemical processes driving the different chemical schemes.

Figure 4 and Table 3 reports the comparison between modeled and observed meteorological parameters at the SMEAR-II station. The site was characterized by rather warm temperatures during the very beginning of the simulations, which later transitioned into a colder period persisting until about 15 July. Afterwards, a sustained increased in temperatures occurred between around the 18 and 29 of July (with daytime temperatures well above 20º) after which the temperatures dropped again until the end of the period. The model was able to reproduce such a temporal trend with a slight underestimation occurring mainly during the nighttime periods (Figure 4). A comparison between modeled and observed relative humidity also indicated



similar level of agreement with the diurnal variation well captured in the model, but some sporadic rain events were missed in the model.

The analysis of the wind direction fields indicated that they were satisfactorily reproduced by the model, with the southern westerly (SW) sector being the most predominant wind direction during the summer period, but with wind speed generally

over predicted. Quite low wind speed values (i.e. around 1 - 1.5 m s$^{-1}$) were observed during the middle of simulation (from the 18 to the 29 of July) concurrently with "heat wave" episode, whereas values were generally higher (2 - 2.5 m s$^{-1}$) during the first half and second half of the investigated periods, a pattern that the model well reproduced (r = 0.62).

### 4.2 Analysis of biogenic volatile organic compounds (BVOCs)

We report here the analysis of the biogenic emissions fluxes (i.e. output from the MEGAN model) for monoterpenes, isoprene, and sesquiterpenes (humulene lumped class) emissions as well as the comparison of their corresponding air concentrations (i.e. output from the CHIMERE model). Figure 5 shows the average spatial distribution of monoterpenes and isoprene BVOCs for the investigated period and for the 10 km resolution nested domain centered over Finland. As expected, monoterpenes emissions clearly dominates the biogenic emission flux over isoprene with a north to south gradient. The model indicated few

localized areas, mainly in the eastern regions of the domain, where substantial isoprene emissions are evident. Specifically, at the location of the SMEAR-II station, isoprene emissions show a larger diurnal variability compared to monoterpenes (Figure 6) with model predicting up to about 59, 36 and 5 % of monoterpenes, isoprene and sesquiterpenes (humulene) relative contributions to the total BVOCs pool, respectively (Figure 7), which is generally in line with previous measurement performed at the same site (Hellén et al., 2018), with the exception of isoprene which shows larger contribution in the model calculations.

Previous model estimation of biogenic emissions over the Finnish forest indicated than monoterpene emissions dominates the total BVOC pool, representing up to about 45 % of the annual total emission whereas isoprene emissions contribute by about 7 % (Lindfors and Laurila, 2000), which is considerably lower than the relative contribution reported in our study here. Those discrepancies can likely arise from the different EFs, and land use types, that are used to retrieve the biogenic emission fluxes. Even though the underlying emission mechanism is very similar compared to the one used in previous studies (Guenther,

1997), the emission factors applied to those early estimates of biogenic flux were specifically retrieved for boreal tree species, differently from the one used here which are directly taken from measurements available in north America and Central Europe. Additionally, in those previous studies, tree species with no document isoprene emission were assigned a minimum emission rate, whereas in this work we applied EFs as implemented in the MEGAN modeling framework (Guenther et al., 2006).

The comparisons of isoprene and monoterpenes air concentrations at the SMEAR-II station is reported in Figure 8. The model

could reproduce relatively well the concentrations of monoterpenes with increasing values occurring during the warmer periods of the investigated period (denoted here as "heat wave") and relative lower values during the colder periods (see Section 4.1). Few isolated spikes in monoterpenes air concentrations are likely to arise from local anthropogenic activities in the nearby sawmill facilities (Heikkinen et al., 2021; Hellén et al., 2018; Vestenius et al., 2021), a feature that is not included in the



emission model. Isoprene concentrations also indicated an increase in concentrations during periods characterized by warmer

temperatures, but concentrations were largely overestimated, which are likely to arise from an over prediction in the biogenic

emissions, as detailed above. Overestimations of isoprene in CTM application with the MEGAN model at the SMEAR-II were

also reported in the study of Jiang et al., 2019a, which used the Comprehensive Air Quality Model with Extensions (CAMx)

model to simulate  the entire year of 2011, and, more recently, in a WRF-CHIMERE application over the pine forest in south-

western France (Cholakian et al., 2022). Even though the model uses a different chemical scheme to perform the gas-phase

and particle phase chemistry, they both indicated a large overestimation of isoprene concentrations, also at other European

sites. The implications of such an overestimation in the biogenic emissions model is analyzed in detail in Section 4.4.

### 4.3 Analysis and source apportionment of OA

The modeled total OA fraction is compared against OA measurements performed with the ACSM instrumentation (Heikkinen

et al., 2021). In the model, this fraction represents the sum of POA (i.e. primary emitted organic material) and SOA (i.e.

secondary formed organic material upon oxidation and subsequently condensation of the resulting low-volatile vapors) from

all the sources considered in the simulation (biogenic, anthropogenic as well as boundary conditions). Figure 9 reports the

hourly and diurnal comparisons for all the three BSOA schemes evaluated. Specifically, the model can reproduce the temporal

trends of the observed OA fraction relatively well: the three main peaks occurring during the beginning, the "heat wave" and

the last week of the investigated periods are all captured, but their magnitude highly depends on the specific BSOA scheme

(Table 1). Similar behaviors were also evident from the comparison of the modeled total $PM_{2.5}$ concentrations against AQ e-

Reporting measurement data (Figure S1 and Table S2).

Additionally, periods with relative low concentrations are also well reproduced, with no substantial positive bias observable.

The analysis of the diurnal profiles indicates that the model can reproduce the daily variation, with a rather flat diurnal of OA

concentrations which increase slightly during nighttime and early morning hours. Extremely low OA concentrations are missed

by the model, and there is a tendency of zeroing out such concentrations throughout the entire simulations (Figure 9 and Figure

10). The latest might suggest uncertainties in the background OA fields used in the model and/or in the concentrations injected

at the very boundaries of the coarser domain (i.e., long-range transport).

The model-based source apportionment of the OA fraction for the three different BSOA schemes is reported in Figure 11 and

Figure 12 for the entire domain as well as for the SMEAR II station. As expected, not much of a difference it is noticed for the

POA and ASOA concentration for the different aging scheme, with POA contributing largely over the urban areas of Helsinki,

Turku, and over the city Tallinn and San Petersburg, with average concentration up to about 0.5 µg m$^{-3}$ on average. The model-

based source apportionment predicted ASOA concentrations to exceeded POA ones, also in urban areas, like for example over

the urban area of Helsinki, and in general, in the southern part of the domain where concentrations are relatively higher

compared to northern regions (up to about 1.5 µg m$^{-3}$). BSOA concentrations, on the other side, were predicted to be more

heterogeneously distributed within the domain (inner domain), and the OA mass largely increases as aging processes are





increasingly accounted for. For instance, in the Aging-Off scenarios, concentrations reach a maximum of around 1 µg m$^{-3}$ on average over the whole period, whereas in the Aging-On-Case-1 case they reach up to 3 µg m$^{-3}$ on average. This represents a substantial difference in the modeled BSOA mass, which is mainly driven by periods characterized by higher temperatures and therefore higher photochemical activity (Figure 9).


The pie chart in Figure 12 reports the modeled average relative contribution of the OA fraction at the SMEAR-II site. Each of the three parameterization indicated that the secondary fraction of OA is the dominant one, which is also in agreement with the positive matrix factorization (PMF) analysis performed on the ACSM data (Heikkinen et al., 2021). In the latter, a statistical source apportionment study of the OA measurement data was performed using the spectral profiles from the ACSM

instrumentation. The authors were able to identify three categories of OA: low-volatility oxygenated OA (LV-OOA), semi-volatile oxygenated OA (SV-OOA), and primary OA (POA). Their results indicated that LV-OOA and SV-OOA almost accounted for the entire OA mass during the summer periods (and eventually also during winter periods), with LV-OOA being the dominant component throughout the entire year. On the other side, the highest SV-OOA contribution to the OA mass was identify during summer periods (about 40%) with a distinct diurnal cycle with peaks in the early morning and in the late

evening (in line with the diurnal profiles indicated by WRF-CHIMERE, Figure 9). Nevertheless, comparison of the PMF-retrieved SV-OOA and LV-OOA components against WRF-CHIMERE data is currently challenging because of the limited number of volatility bins used in the model to describe the formation of BSOA (i.e. currently limited to 4 at 1, 10, 100, and 1000 µg m$^{-3}$ at 300 K), which only partially cover the low-volatile range (Bianchi et al., 2019). This limitation was further investigated by comparing the model size distribution with DMSP measurements at the SMEAR-II site. The model largely

overestimates the number of particles below 100 nanometers and underestimate in the accumulation mode (Figure S2). These behavior were already reported in the work of Tuccella et al., 2019 which used aircraft measurement data (available both in the PBL and in the free-troposphere) to evaluate the model size distribution. Whether we acknowledge that this model version does not account for any adjustment of the organic compounds based on their size, i.e. Kelvin effect (which will reduce the amount of semi-volatile compounds condensing on very small particles size, mainly below the 10 nanometers sizes), and that

the number of particles is retrieved in a prognostic manner from the total OA mass, density, and particle diameter, it is likely that the lack of a more explicit representation of the LVOC and ELVOC compounds (i.e. volatility bins) in this VBS framework (Figure 2) could potentially reduce the growth, by condensation, of particle in the lower end of the size distribution, therefore adding to the overestimation in the smaller diameters and to the under prediction of the larger ones (which will additional reduce the coagulation efficiency of smaller particles towards larger sizes).

Finally, the model-based relative contribution of ASOA and BSOA to the total OA mass indicates substantial variations depending on the adopted chemical scheme. In particular, for the Aging-Off test the model predicted up to 43 % contribution of the anthropogenic SOA fraction to the total OA mass, which is very likely overestimated for the SMEAR-II boreal site. The aging of BSOA yield results that are more reasonable both in the terms of the contribution of the single OA components and also in the terms of the absolute concentrations (for the Aging-On-Case-2 BSOA scheme), i.e. with the BSOA fraction

contributing up to 72 % to the total OA and the total OA mass negatively biased by 0.7 µg m$^{-3}$ (Table 5).





### 4.4 Sensitivity of BSOA formation to $C_5H_8$ emissions

We discuss in this section sensitivity analysis with inhibited isoprene biogenic emissions. As presented in the previous section, modeled isoprene air concentrations were largely overestimated at the SMEAR-II site (likely because of the overstated isoprene
emissions) and especially during periods characterized by high temperatures and intensive photochemical activity (referred to as "heat wave" episode). Particle mass yield from isoprene biogenic compounds is lower compared to monoterpenes and recent studies have shown that isoprene can effectively scavenge $^{\cdot}$OH radicals, preventing their reactions against other terpenoids therefore limiting the formation of biogenic aerosol particles and the total organic mass (McFiggans et al., 2019).

Figure 13 reports the daytime relative changes in α-pinene, $O_3$ and BSOA concentrations between the two simulations
performed with and without isoprene emissions. Inhibiting isoprene emissions resulted in a non-negligible increased in the BSOA mass concentrations almost all over the high-resolution nested domain (i.e. Finland). In most of the areas, the BSOA mass increased by about 10 % with maximum increases at around 25 % especially in the southern part of the domain and over few regions of the Baltic sea. Conversely, α-pinene air concentrations were homogenously reduced all over the domain (Figure 13). The relative reductions (over land) were in the order of 10 to 20 %. As isoprene emissions are excluded from the modelling
system, more α-pinene of biogenic origin can effectively reacts towards available radicals, i.e. $^{\cdot}$OH radicals, and, owing to its higher mass yield compared to isoprene (Hodzic and Jimenez, 2011), effectively increase the production efficiency of BSOA. Comparisons of daytime OA aerosol mass against ACSM data described in the previous section, additionally indicated a slight improvement in model's performance, especially during periods characterized by elevated temperatures (Figure 14). These non-linear response to isoprene emissions, seems to have also important effect on the long-range transport of the BSOA mass.
Largest increases are indeed predicted over water bodies (with zero biogenic emissions and low deposition efficiency), suggesting that also the formation of low volatile and aged organic aerosol masses might be likewise affected. Figure 13 also reported the relative changes in $O_3$ concentrations between the two simulations performed with and without isoprene emissions which were predicted to be very mild especially over the inland areas of the domain. Formation of $O_3$ is driven by the availability of both $NO_x$ and VOCs emissions, with the investigated area clearly belonging to a $NO_x$-limited regime. As
reported in the Figure 15, the model is capable to reproduce the diurnal variation and absolute values (ppb) of $O_3$ very well, whereas $NO_x$ concentrations were overestimated during nighttime periods, a behavior that can be induced by a too shallow planetary boundary layer (PBL) in the model. Additionally, both model and measurements data, did not show a substantial local production of $O_3$ concentrations, i.e., both model and observational $O_3$ values increased from about 25 to about 30 ppb during daytime. As a comparison, measurements data in the Po Valley (an heavily polluted region often experiencing high
$NO_x$ concentrations) indicated that the production of $O_3$ can range from 25 to 50 ppb, even in colder periods of the years, i.e. March and April (Ciarelli et al., 2021). This suggests that a large fraction of the $O_3$ measured at the SMEAR-II sites is of long-range origin, therefore explaining the relative low changes in its concentrations between the two scenarios, as also reported by previous studies (Curci et al., 2009).



## 4.4 Impacts on cloud liquid water content (clwc)

We report a preliminary analysis for the changes in modeled Cloud Liquid Water Content (clwc) between the different aging schemes. As discussed in the Section 2.1, we run the model in "online" configuration, therefore allowing CHIMERE to pass the diagnosed particle size number distribution, aerosol bulk hygroscopicity, ice nuclei (IN) and deliquesced aerosol to the WRF model. CHIMERE chemical and physical parameters were passed to the WRF model with an exchange frequency of 20 minutes and the aerosol activation to cloud droplets treated with the Abdul-Razzak and Ghan scheme (Abdul-Razzak, 2002) using a similar approach available in the WRF-Chem model (Chapman et al., 2009). More detailed on the Aerosol-Cloud Interaction within WRF-CHIMERE can be found in (Tuccella et al., 2019).

Figure 16 reports the vertically integrated average relative changes in clwc between the Aging-On-Case-2 and Aging-Off schemes for the entire simulated period. We calculated these differences using the Aging-On-Case-2 scenarios since, the latest, yields the best results against OA measurements (Section 4.3). We reported the vertical distribution of aerosol particles up to 150 nanometers, i.e. including particles that can effectively act as could condensation nuclei (CCN), as in the Aging-On-Case-2 and Aging-Off cases. Generally, the model indicated an increase in the clwc when the aging of biogenic aerosol are accounted for, owing to the increase in the biogenic aerosol mass loading (Figure 10) and therefore of the number of biogenic particles that can act as CCN (Figure S2). Changes in clwc are predicted to be larger over the land, and especially in the central region of the domain and over the SMEAR-II site where the model indicates around 30% increase in clwc in the Aging-On-Case-2 respect to the Aging-Off. Most of the changes, i.e. total number of particles up to 150 nanometers and clwc, occurred below about 1000 meters a.s.l. which is roughly the estimated average PBL height during summer period at SMEAR-II (Sinclair et al., 2022), with larger increases, in both particles number and clwc, slight below the 1000 m a.s.l altitude. Climatic feedbacks from biogenic particles over boreal area were very recently reported in Yli-Juuti et al., 2021 by the means of remote sensing and ACSM observations available at the SMEAR-II stations. Specifically, the analysis indicated an increase in OA loading, and CCN, during the 2012 - 2018 periods as results of the increase in surface temperature. Higher cloud optical depth data were also statistically-significant associated with higher OA loading, providing direct evidence for the indirect effect of biogenic aerosols. The model results presented here, seem to be in line with such results, but a separate study is needed to analyze in greater detail the modeled indirect effect under longer periods of time (i.e., model trend analysis).



## 4 Conclusions

We presented a modeling evaluation study aiming at evaluating the formation of biogenic secondary organic aerosol over the Finnish Boreal Forest with the WRF-CHIMEREv2020r2 model. We investigated the formation of BSOA using different aging schemes to treat the second-generation oxidation product of BSOA, also referred to as chemical aging, as currently available

from literature. Results were evaluated against high-resolution organic aerosol (OA) measurements performed with an aerosol chemical speciation monitor (ACSM) at the SMEAR-II site, an area largely affected by biogenic emissions. We used parallel measurements of biogenic gas-phase precursors (i.e., isoprene and monoterpenes) to investigate the model performance with respect to the BSOA precursors which offers a proper framework to evaluate Chemical Transport Model (CTM) simulations to a greater level of details. Additionally, we evaluate the model's respond to changes in isoprene emissions and the impact of

different chemical scheme on the predicted Cloud Liquid Water Content (clwc).

The meteorological evaluation of standard parameters affecting the formation and transportation of BSOA was found be to satisfactory reproduced throughout the whole simulated period, underlying the capability of the WRF model to properly reproduce the meteorological regimes of the summer of 2019 on a 10 kilometers grid resolution.

The model could well reproduce the diurnal variation of the OA mass as measured at the SMEAR-II site. As expected, aging

processes of BSOA largely increased the BSOA mass, yielding reasonable model performance, both in term of the total OA mass as well as in terms of sources contribution (i.e. POA, ASOA and BSOA), for schemes that account for aging of BSOA as proposed in previous study (Bergström et al., 2012). On the other side, the analysis of the model size distribution indicated a large overestimation for particles below about 100 nanometers and an underestimation for particles in the larger diameter sizes. We attributed such compensating effect to 1) a lack of an explicit treatment of organic compounds in the LVOC and

ELVOC range, which can effectively condense on smaller particles and promote their growth to larger particles size, and to a lesser extent 2) to the lack of the kelvin effect in the model calculation. Overall, those results stressed once more the need to proper represent the volatility distribution in CTM application, and more work is needed towards this direction, particularly on the volatility distribution of BSOA and on the implementation of physical processes affecting the evolution of the size distribution (e.g., inclusion of organic gases in the very low volatility ranges).

Additionally, the analysis of biogenic gas-phase precursors indicated that the model largely overestimated isoprene emissions likely because of overstated emissions in the MEGAN emission model, as already presented in various application of the MEGAN model at European scale. There is still a need to further reduce the uncertainties in the current estimation of biogenic fluxes, one of the key parameters influencing the formation of BSOA in CTMs. An additional sensitive test indicated that an overestimation could potentially reduce the production of BSOA by scavenging away ˙OH radicals that would have been

available to react against α-pinene compounds which have higher SOA yields.

Finally, our model results indicated an increase in the cloud liquid water content when aging of BSOA are accounted for, likely because of the increased number of aerosols particles acting as could condensation nuclei (CCN), as recently also suggested





by measurements studies conducted at the SMEAR-II site. These preliminary results reported here should be corroborated in a greater level of details by a more detailed modeling studying spanning multiple years (i.e., trend-analysis).

Overall, the model evaluation presented here indicated once more the importance of properly characterize both biogenic emissions fluxes and chemical scheme parameterization to correctly predict the formation of BSOA and its size distribution. As climate continues to warm, biogenic emissions could become an increasingly important contributor to the total OA pool, and model predictions could largely varies depending on the level of confidence of emission strength and chemical parameterization.




**Code availability**

The WRF-CHIMERE model is freely available at https://www.lmd.polytechnique.fr/chimere/ (registration required).

**Data availability.**

The input data used for the simulations are available at https://www.lmd.polytechnique.fr/chimere/2020_getcode.php. The
CAM anthropogenic emission can be downloaded at https://eccad3.sedoo.fr/catalogue, dataset name: CAMS-GLOB-ANT.
Measurements data from the SMEAR-II station can be downloaded at https://smear.avaa.csc.fi/download. The ACSM NR-
PM1 OA concentration data are available on the EBAS database under the EMEP ACTRIS framework (http://ebas.nilu.no/).

**Acknowledgment**

This work was supported by the European Research Council via the project CHAPAs (No. 850614) and by the Academy of
Finland (Nos. 311932, 307537, 334792, 337549). Model simulations were performed "online" with the meteorological model
on the mahti supercomputer of the Finnish IT center for science (CSC) using 1 computing node for the European grid, and 4
computing nodes for the high-resolution domain. We would like to thank Juha Lento for his continuous support at the Finnish
IT center for science (CSC). We are also thankful to Liine Heikkinen and Mikael Ehn for their inputs on the manuscript.

**Author contributions**

G. C. designed and led the work in collaboration with F.B., performed the WRF-CHIMERE simulations, and wrote the paper.
A.C. provided technical support during the operational phases of the WRF-CHIMERE model, recommendations on the model
set-up and errors handling support. S. T. performed the data analysis of the model data in collaboration with B.V.. T.P. provided
technical support in handling the instrumentation datasets. All co-authors review, commented and supported the interpretation
of the results presented in the paper.

**Conflicts of interest**

The authors declare that they have no conflict of interest.





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

**Figures and Tables**

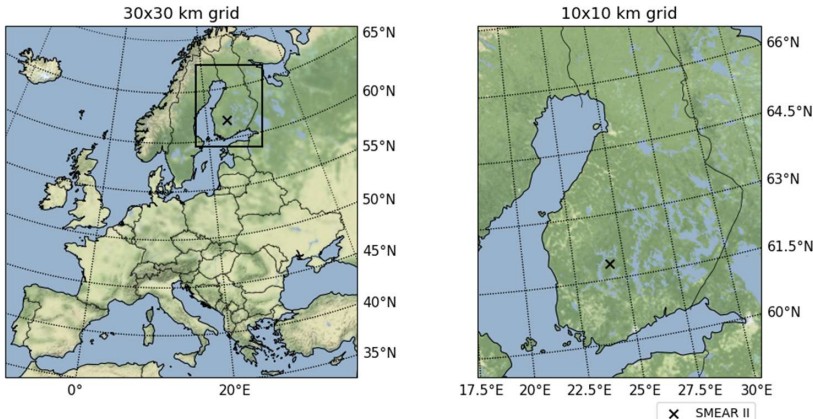

**Figure 1: The two model domains: the European grid (left) with a cell size of ~ 30 x 30 km, and the nested grid (right) with a cell size of ~ 10 x 10 km (right). The black square on the European grid (left) indicates the positon of the nesting. The black cross denotes the location of the SMEAR-II station.**


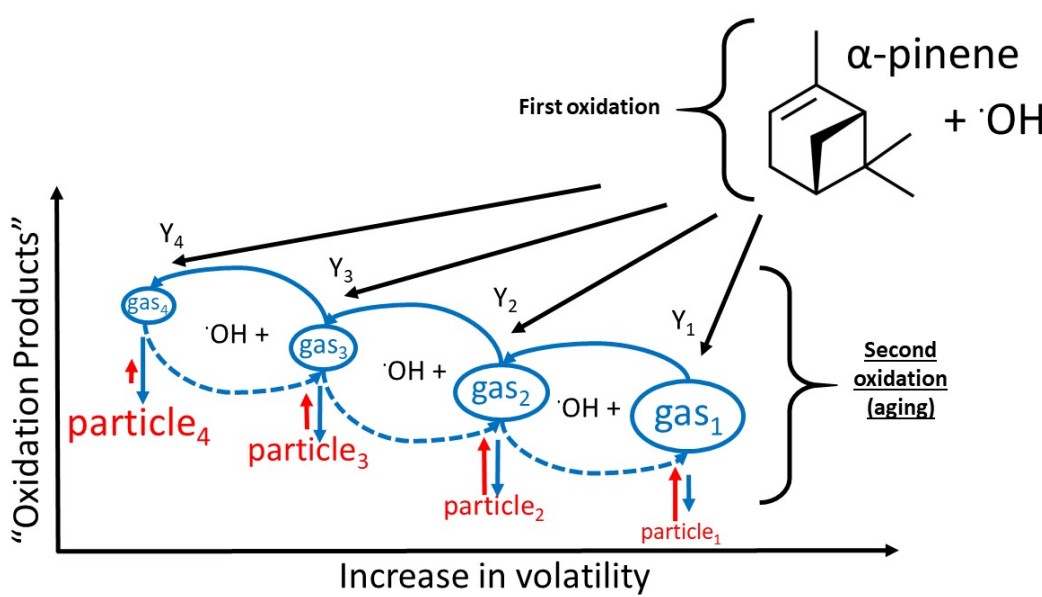

**Figure 2: Schematic of the oxidation scheme of biogenic precursors as implemented in the VBS scheme of CHIMERE (here reported specifically for the $C_{10}H_{16}$ parent precursor). The black arrows represent the distribution kernel of the first oxidation products into the four volatility bins ($Y_1…Y_4$). The blue curved arrows represent the secondary oxidation processes, i.e. aging, along the four volatility bins, each of which decreases the volatility by one order of magnitude. The text font size represents the tendency of both particles and gas-phase organic material (OM) to transition in the one or the other phase (i.e. larger font size indicated a better attitude towards that phase, and vice versa). The dashed curved arrows represent the fragmentation process (also available in CHIMERE, but not currently used for this application).**



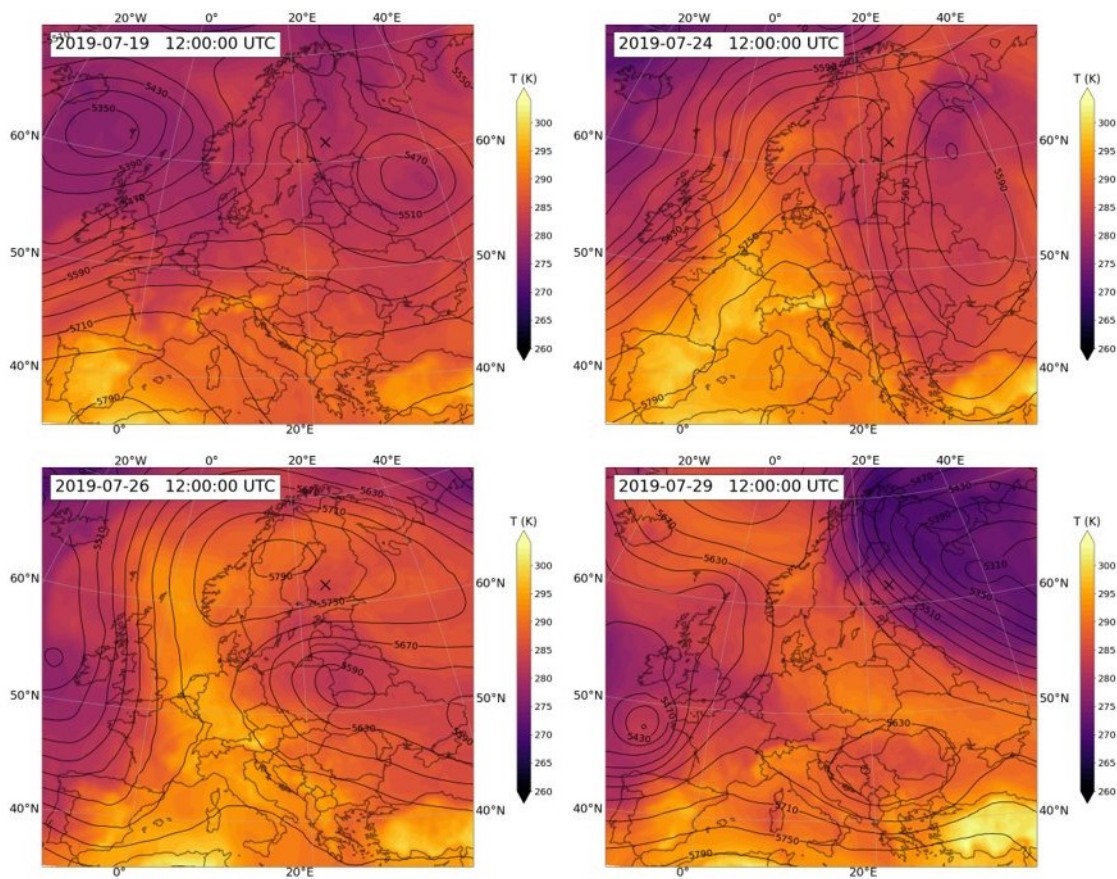

**Figure 3: Geopotential height (m a.s.l.) at 500 hPa and air temperature (K) at 850 hPa for 4 days during the heat wave period (19-28 July). Data are taken from ERA5 reanalysis (available at cds.climate.copernicus.eu). The black cross denotes the location of the SMEAR-II station.**


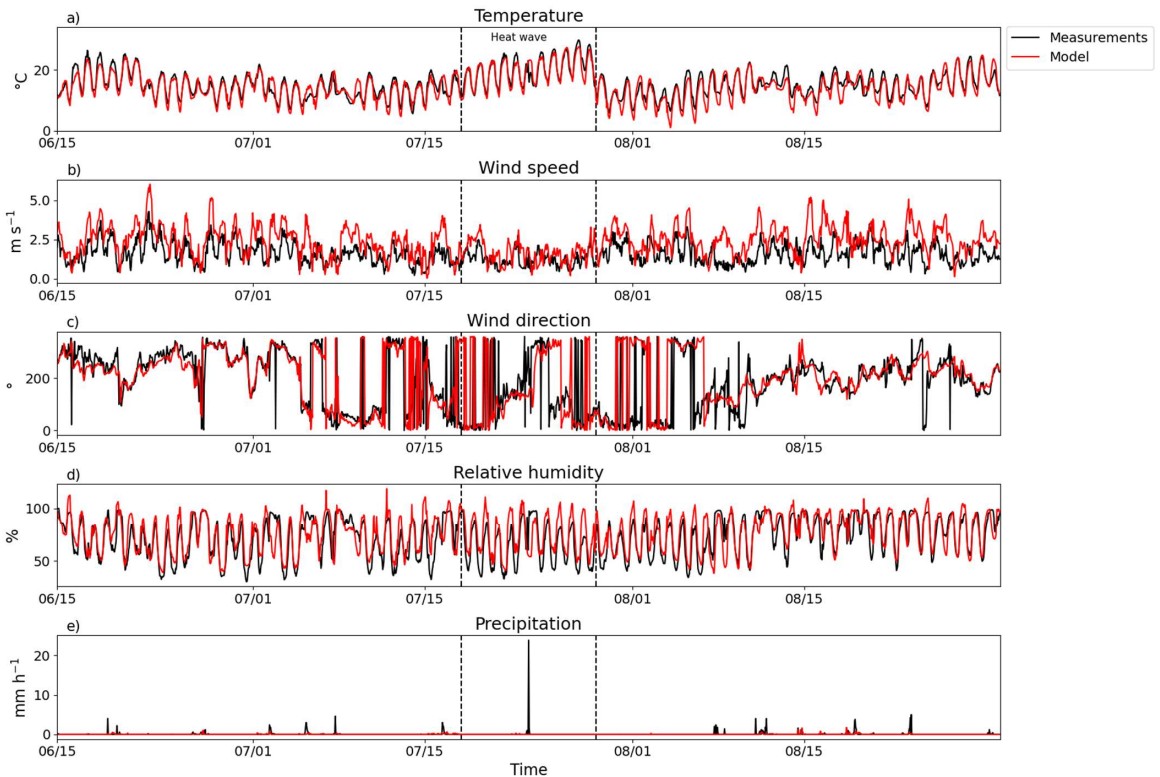

**Figure 4: Hourly comparison of different meteorological parameters at the SMEAR-II station. From the top to the bottom: a)**
**temperature (°C), b) wind speed (m s⁻¹), c) wind direction (°), d) relative humidity (%) and e) precipitation (mm h⁻¹). Black lines**
**indicates the measurement data and red lines the model data. The dashed lines delimits the periods with sustained elevated**
**temperature, denote here as "heat wave".**



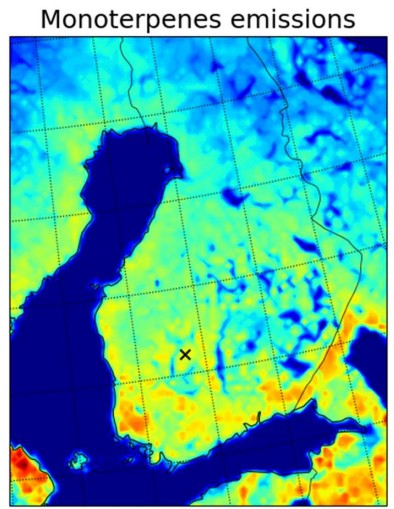
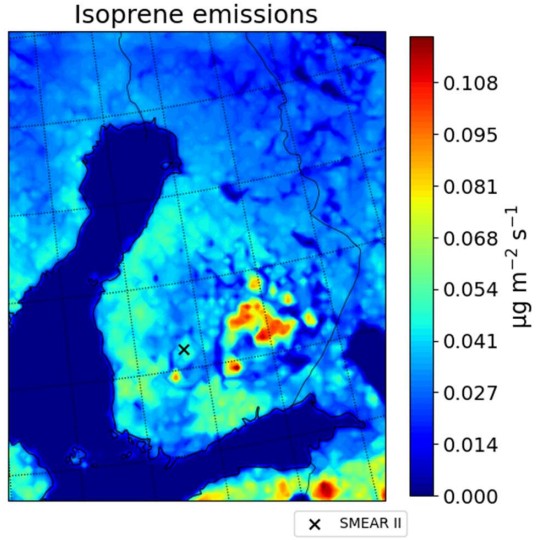

**Figure 5: Average spatial distribution of monoterpenes (left) and isoprene (right) emissions ($\mu$g m$^{-2}$ s$^{-1}$) for the astronomical summer of 2019. The cross denotes the location of the SMEAR-II station. Monoterpenes represent here the sum of all the available terpenes species.**



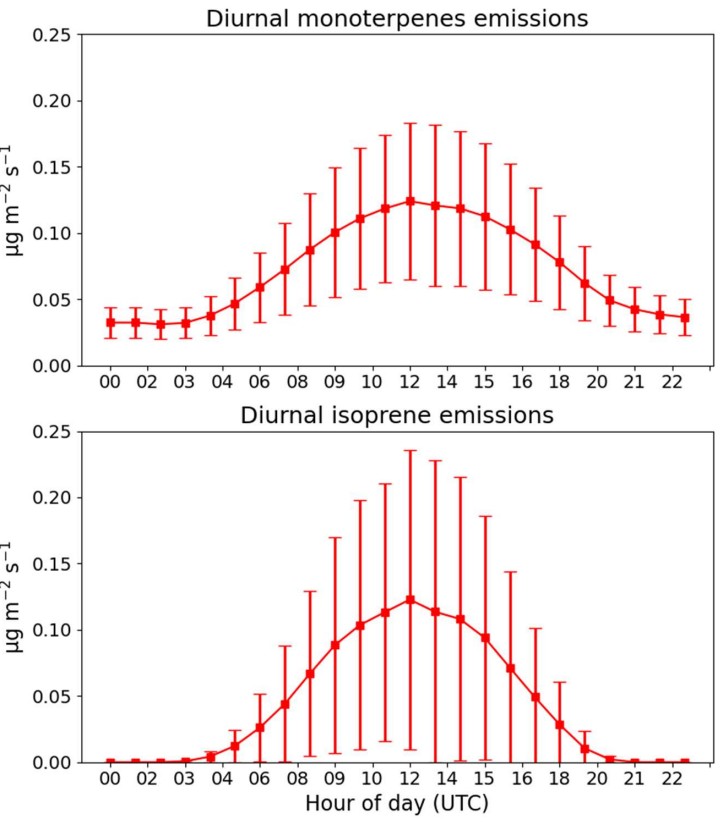

**Figure 6: Average diurnal variation of monoterpenes (upper panel) and isoprene (lower panel) emissions ($\mu$g m$^{-2}$ s) for the**
**astronomical summer of 2019 at the SMEAR-II station. The extent of the red bars denotes the one standard deviation (1$\sigma$). Monoterpenes represent here the sum of all the available terpenes species.**

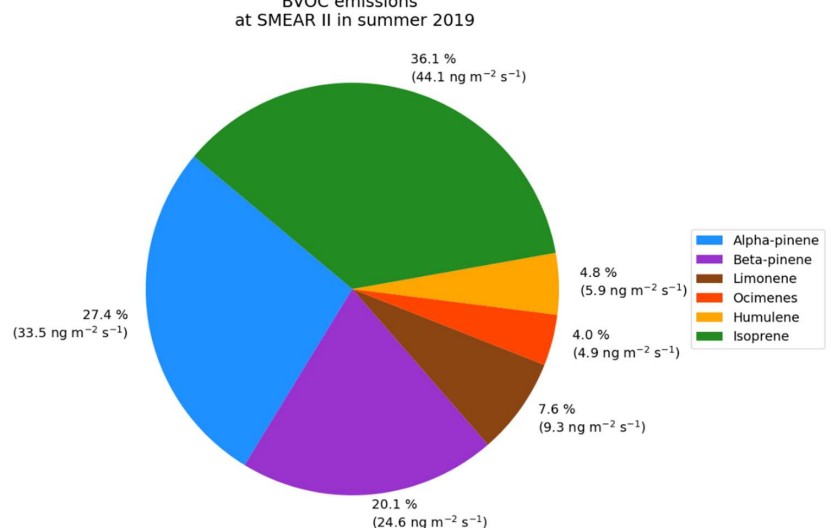

**Figure 7: Average relative contribution of the BVOCs species as predicted by the MEGAN model for the astronomical summer of 2019 at the SMEAR-II station. Units are in ng m$^{-2}$ s$^{-1}$.**






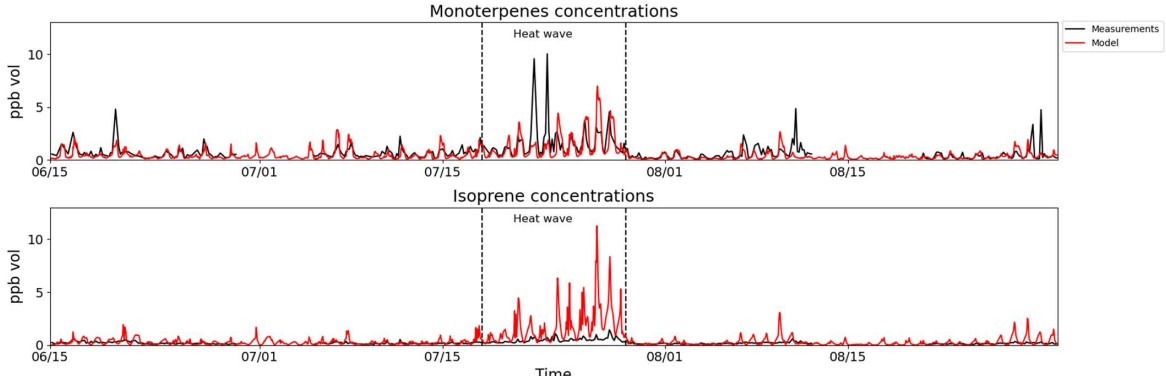

**Figure 8: Hourly comparisons of model (red) and measured (black) air concentrations of a) isoprene and b) monoterpenes (sum of terpenes) at the SMEAR-II station. Units are in ppb vol. The dashed lines delimits the periods with sustained elevated temperature, denote here as "heat wave".**


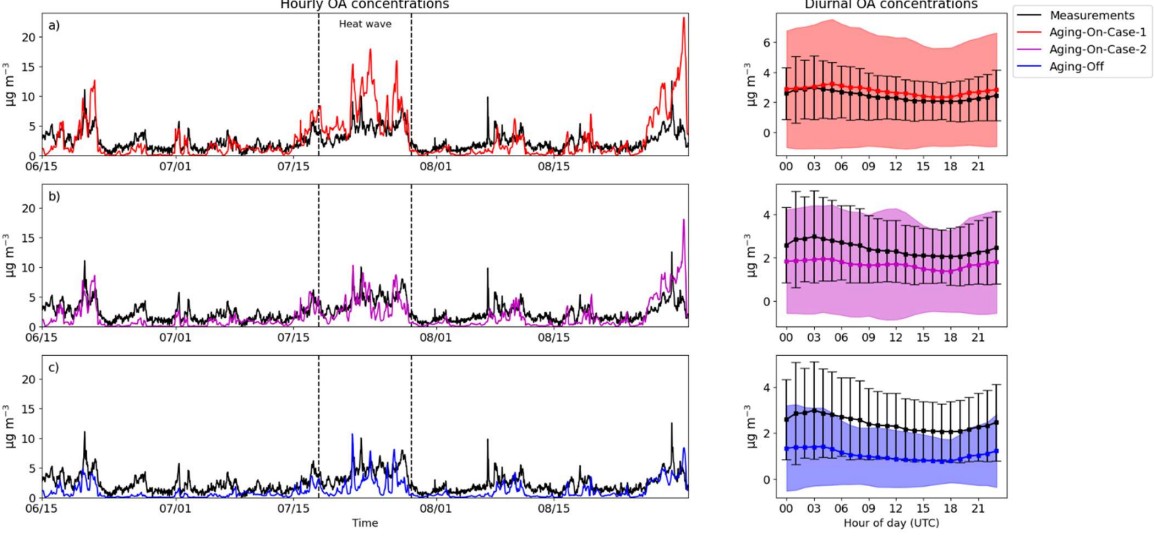

**Figure 9: Model (different colors) and measured (black) air concentrations of OA for the a) Aging-On-Case-1, b) Aging-On-Case-2 and c) Aging-Off BSOA schemes. Hourly (left) and diurnal (right) comparisons at the SMEAR-II station. The dashed lines delimits the periods with sustained elevated temperature, denote here as "heat wave". The extent of the bars and the shaded areas denotes the one standard deviation (1σ). Units are in μg m⁻³.**


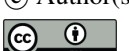

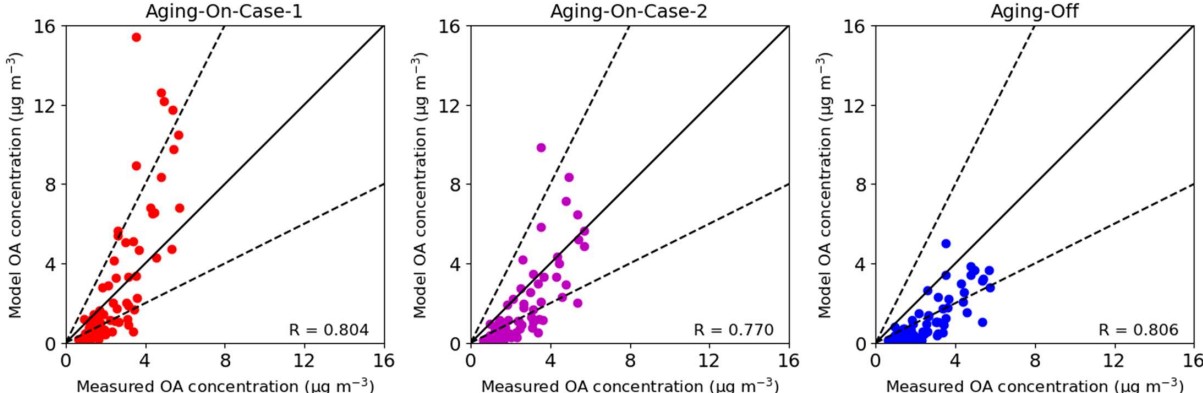

**Figure 10: Daily average model (y-axis) and measured (x-axis) air concentrations of OA for the Aging-On-Case-1 (left), b) Aging-On-Case-2 (center) and c) Aging-Off (right) BSOA schemes at the SMEAR-II station. Solid line indicates the 1:1 line. The dashed lines delimits 1:2 and 2:1 lines. Units are in µg m⁻³.**





**Figure 11: POA (top panel), ASOA (middle panel) and BSOA (bottom panel) average concentrations during the astronomical summer of 2019 and for the Aging-On-Case-1 (left), Aging-On-Case-2 (center) and the Aging-Off (right) BSOA schemes. The cross denotes the location of the SMEAR-II station. Units are in µg m⁻³. A different scale in used for the BSOA panel (bottom panel) in order to facilitate the comprehensions of the panel.**






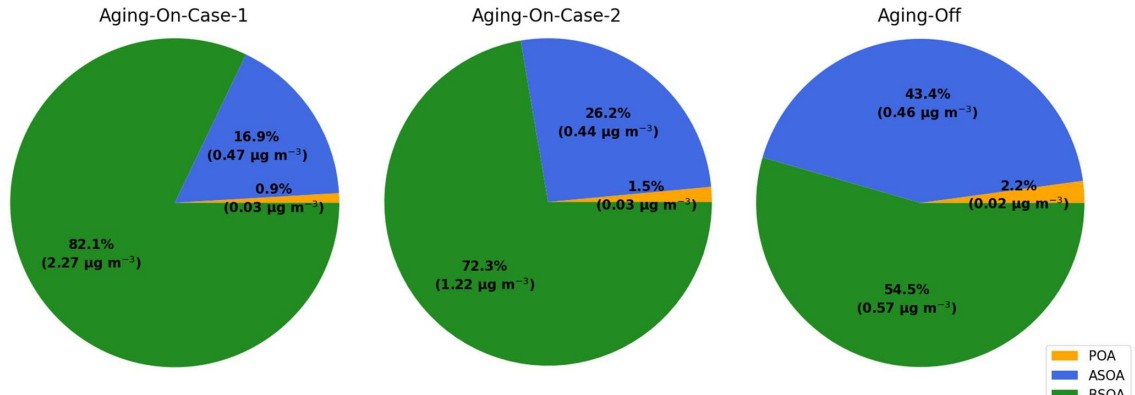

**Figure 12: POA (orange), ASOA (blue) and BSOA (green) modeled average relative contribution to the total OA fraction for the astronomical summer of 2019 and for the Aging-On-Case-1 (left), b) Aging-On-Case-2 (center) and c) Aging-Off (right) BSOA schemes. Absolute concentration are reported along with their relative contribution to the total OA.**



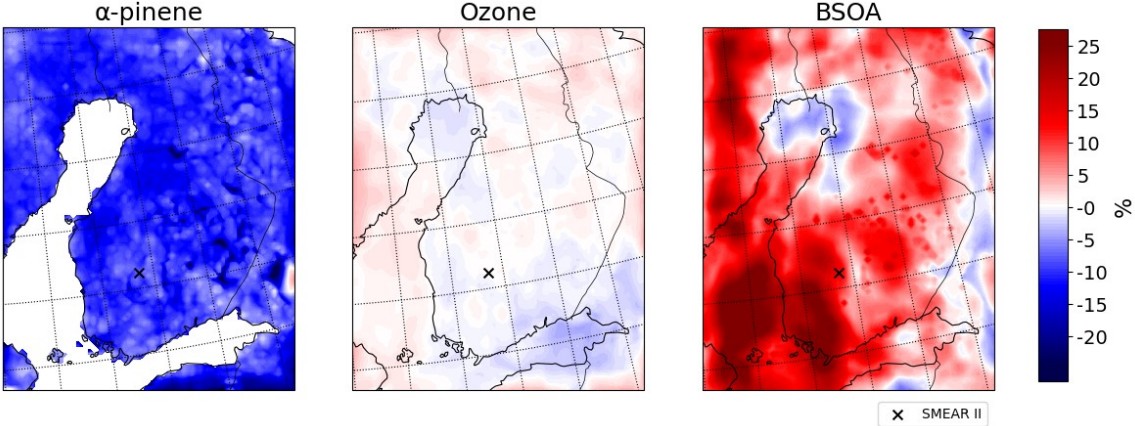

**Figure 13: Daytime average (08 - 20 LT) relative changes in $C_{10}H_{16}$ (alpha-pinene) air concentrations, Ozone and BSOA concentrations with and without isoprene emissions. The relative changes are calculated as (($C_5H_8$-emissions-Off - Aging-On-Case-2) / Aging-On-Case-2) * 100.**



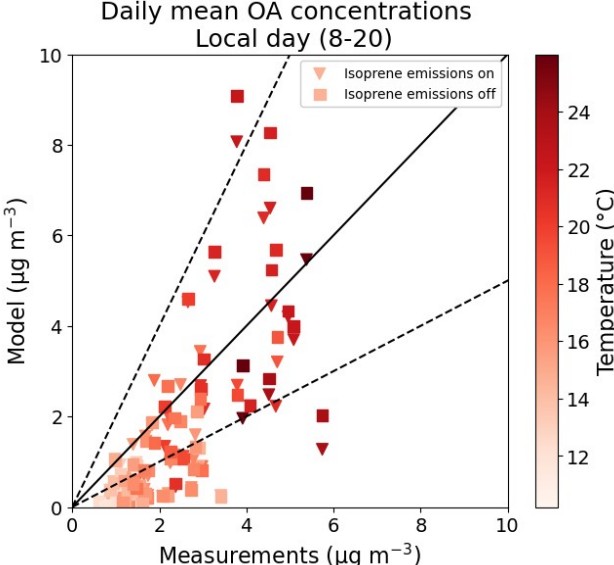

**Figure 14: Daytime average (08 - 20 LT) model (y-axis) and measurements (x-axis) daily comparisons of OA mass at the SMEAR-II station as a function of temperature with isoprene emissions activate (triangles) and isoprene emissions inhibited (squares). Units are in μg m⁻³.**





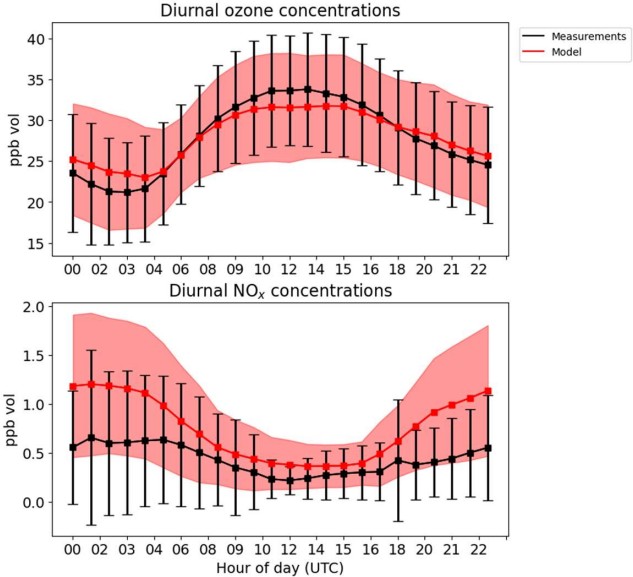

**Figure 15: Diurnal variation of O₃ and NOₓ at the SMEAR-II site (from 15 June until 30 August of 2019). The extent of the bars and the shaded areas denotes the one standard deviation (1σ). Measurements data are shown in in black and model data in red. Units are in ppb vol.**



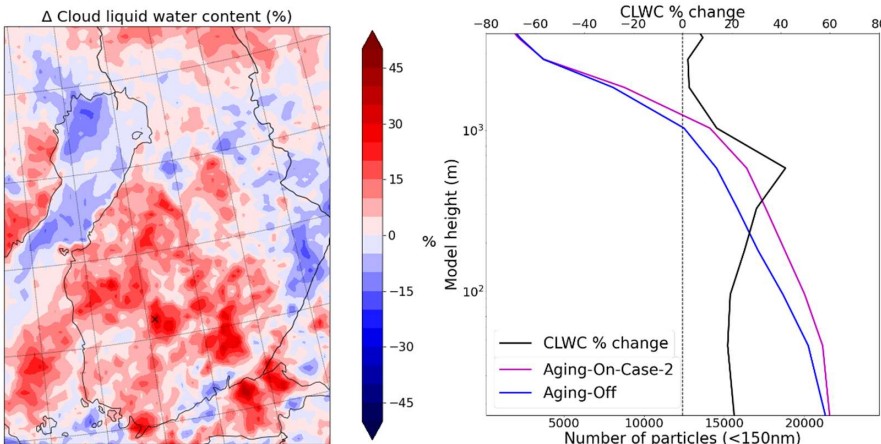


**Figure 16: Integrated (vertical) differences in Cloud Liquid Water Content (left) over the high-resolution domain (10km). Vertical profile of particles below 150 nanometers in the Aging-On-Case-2 and Aging-Off simulations and of the relative changes in Cloud Liquid Water Content (right) over the SMEAR-II site (average of a 3x3 kilometers cell). The relative changes are reported here as 840 ((Aging-On-Case-2 - Aging-Off) / Aging-Off) * 100.**



**Table 1: Reaction rates constant of BSOA aging as used in the different sensitivity tests. The schematic of the aging scheme is reported in Figure 2.**


| Sensitivity Test | Aging (see Figure 2) |
|---|---|
| Aging-On-Case-1 | $1 \times 10^{-11}$ molecule$^{-1}$ cm$^3$ s$^{-1}$ |
| Aging-On-Case-2 | $4 \times 10^{-12}$ molecule$^{-1}$ cm$^3$ s$^{-1}$ |
| Aging-Off | $0$ molecule$^{-1}$ cm$^3$ s$^{-1}$ |
| $C_5H_8$-emissions-Off | $4 \times 10^{-12}$ molecule$^{-1}$ cm$^3$ s$^{-1}$ |

**Table 2: Statistical metrics used for model evaluation. $M_i$ and $O_i$ stand for modeled and observed values, respectively, and $N$ is the total number of paired values.**

| Metric | Definition |
|---|---|
| Mean Bias (MB) | $MB = \dfrac{1}{N}\sum_{i=1}^{N}(M_i - O_i)$ |
| Mean Gross Error (MGE) | $ME = \dfrac{1}{N}\sum_{i=1}^{N}|M_i - O_i|$ |
| Root Mean Square Error (RMSE) | $RMSE = \sqrt{\dfrac{1}{N}\sum_{i=1}^{N}(M_i - O_i)^2}$ |
| Index of Agreement (IOA) | $IOA = 1 - \dfrac{N \cdot RMSE^2}{\sum_{i=1}^{N}\left(\left|M_i - \overline{O}\right| + \left|O_i - \overline{O}\right|\right)^2}$ |
| Pearson Correlation Coefficient (r) | $r = \dfrac{\sum_{i=1}^{N}(M_i - \overline{M}) \cdot (O_i - \overline{O})}{\sqrt{\dfrac{1}{N}\sum_{i=1}^{N}\left(M_i - \overline{M}\right)^2} \cdot \sqrt{\dfrac{1}{N}\sum_{i=1}^{N}\left(O_i - \overline{O}\right)^2}}$ |
| Mean Fractional Bias (MFB) | $MFB = \dfrac{1}{N}\sum_{i=1}^{N}\dfrac{2 \cdot (M_i - O_i)}{M_i + O_i}$ |
| Mean Fractional Error (MFE) | $MFE = \dfrac{1}{N}\sum_{i=1}^{N}\dfrac{2 \cdot |M_i - O_i|}{M_i + O_i}$ |






**Table 3: Model evaluation for the meteorological parameters (from 15 June until 30 August of 2019). Statistics is performed at 1-hour time resolution.**

| Variable | MB | MGE | RMSE | IOA (-) | r (-) |
|---|---|---|---|---|---|
| T (° C) | -0.7 | 1.6 | 2.0 | 1.0 | 0.9 |
| Wind speed (m s$^{-1}$) | 0.8 | 0.9 | 1.1 | 0.6 | 0.6 |
| Water vapor mixing ratio (g kg$^{-1}$) | 0.2 | 0.9 | 1.1 | 0.9 | 0.8 |

**Table 4: Model evaluation for the BVOCs species isoprene and monoterpenes at the SEMAR II station (from 15 June until 30 August of 2019). Statistics is performed at 1-hour time resolution.**

| Variable | Mean measurements | Mean model | MB | MGE | r (-) |
|---|---|---|---|---|---|
| Monoterpenes (ppb) | 0.8 | 0.6 | -0.2 | 0.4 | 0.5 |
| Isoprene (ppb) | 0.2 | 0.5 | 0.3 | 0.3 | 0.6 |
| O$_3$ (ppb) | 27.9 | 27.9 | -0.1 | 5.6 | 0.5 |
| NO$_x$ (ppb) | 0.4 | 0.8 | 0.3 | 0.5 | 0.4 |

**Table 5: Model evaluation for OA as predicted by the three BSOA schemes at the SEMAR II station (from 15 June until 30 August of 2019). Statistics is performed at 1-hour time resolution.**

| Variable | Mean | MB | MGE | RMSE | MFB | MFE |
|---|---|---|---|---|---|---|
| Aging-On-Case-1 (µg m$^{-3}$) | 2.8 | 0.3 | 1.8 | 2.9 | -0.5 | 0.9 |
| Aging-On-Case-2 (µg m$^{-3}$) | 1.7 | -0.7 | 1.4 | 1.9 | -0.8 | 1.0 |
| Aging-Off (µg m$^{-3}$) | 1.1 | -1.4 | 1.5 | 1.9 | -1.1 | 1.1 |
| Obs (µg m$^{-3}$) | 2.5 | - | - | - | - | - |
