# Peer review of "On the formation of biogenic secondary organic aerosol in chemical transport models: an evaluation of the WRF-CHIMERE (v2020r2) model with a focus over the Finnish boreal forest"

_Geoscientific Model Development, 2023_

## Referee Comment (RC1)

**General comments**

This manuscript presents an evaluation and sensitivity analysis of the WRF-CHIMERE model, with a focus on biogenic secondary organic aerosol and the Hyytiälä monitoring site in Finland. While the topic is within the scope of GMD, the current version of the manuscript is not suitable for publication in the journal. The evaluation of the model is too limited to be of general interest, as it only covers a single site for two and a half months, which is not sufficient for a regional CTM model evaluation paper in GMD.

The paper does not present any substantial novel concepts, ideas, tools, or data, and does not represent a significant advance in modeling science. The short simulation period and apparently poor emission data for isoprene make it difficult to draw firm conclusions from the study, and there are few interpretations or conclusions presented in the manuscript.

If the model evaluation were extended to include more sites across Europe, it could be of sufficient interest to warrant publication in GMD. Organic carbon (OC) measurement data from 2019 are available from ebas.nilu.no for about 30 different regional sites in Europe, which would be a valuable addition to the evaluation. Given the poor agreement with observations for isoprene at Hyytiälä, it would be interesting to include isoprene measurements from other European sites (data from almost 20 sites are available for 2019 in ebas – including data from Pallas in Finland). It would also be useful to investigate the effect of isoprene emissions on ozone across Europe, using observations from the many regional background sites available in ebas. For some unclear reason, the comparison with observed $PM_{2.5}$ measurements was restricted to sites in Spain and Italy (and this comparison is not discussed in detail in the manuscript). It would be interesting to evaluate the model's performance for $PM_{2.5}$ across all of Europe (including Hyytiälä) and discuss these results in the manuscript.

In its current form, I recommend that the manuscript be rejected for publication in GMD. However, if the authors are willing to substantially extend their comparison with observations across Europe, their work may be reconsidered for publication.

**Specific comments**

- Section 2.2 only describes BSOA – how did you treat SOA from anthropogenic VOC? Did you include a VBS-treatment with (or without) aging also for ASOA?

- You base your VBS scheme on Hodzic and Jimenez (2011) – their scheme only included SOA from OH-reactions. Did you include SOA formation from oxidation by ozone and/or $NO_3$ radicals? Please provide details about how the BVOC + $O_3$ and BVOC + $NO_3$ are treated in the model.

- How did you treat SOA from SQT?

- Hodzic and Jimenez (2011) only had a single monoterpene species (TERP) – you split the MT into four different species; please provide details of the differences in SOA yields (and reactivities) for the different MTs.

- Hodzic and Jimenez (2011) also included SOA production from biomass burning, POA ageing, acid-enhanced BSOA production and anthropogenic pollution-enhanced SOA production. Did you include all (or any) of these SOA formation reactions?

- Lines 146–147: You use $\Delta H_{vap}$ of 36 kJ mol$^{-1}$ – Hodzic & Jimenez 2011 used 88 kJ mol$^{-1}$ – please explain why you chose the lower value.

- Did you only include the organic mass (including or excluding particulate water; or the full particle mass) when calculating the gas-particle partitioning of the SVOCs?

- How was the deposition of gas-phase SVOCs treated in the different model simulations?

- Line 165: How were the annual anthropogenic emissions from CAMS "hourly distributed" for the simulation period? Please provide some details (and/or reference) regarding the temporal distribution of the emissions.

- Did you include any emissions from biomass burning (wildfires) in the simulations?

- Lines 179–181: Were the initial and boundary concentrations of aerosols and gases taken from LMDz-INCA3 simulations for 2019 or for some other year? Were the boundary concentrations constant or varying in time?

- Lines 198–204: The selection of $PM_{2.5}$ data for the model evaluation is very odd for a study focussed on the "Finnish boreal forest". Also, there is no discussion of the results of the $PM_{2.5}$ evaluation in the manuscript (only a figure and a table in the Supplement, with no accompanying text). As mentioned in my General comments, I think that the comparisons of modelled and measured $PM_{2.5}$ should be made for all of Europe – and definitely include data from Hyytiälä. The Spanish and Italian sites included here are probably among the least interesting sites for the boreal Finnish forest so unless the $PM_{2.5}$ evaluation is extended I think it should be removed completely.

- Lines 252–254: In what sense are your modelled BVOC emissions "generally in line with" the data presented by Hellén et al. (2018)? I do not think they are very much in line at all. For the summer 2016 they give (in their Table 1) mean total MT concentrations of 427 ppt and only 11 ppt isoprene and 13 ppt SQT; this seems very far from your ratios between the three BVOC types.

- Figure 8 and Table 4. How does the statistics for the BVOCs change if you exclude the sawmill-emission related time periods from the analysis? Does this improve the statistics for MT significantly?

- Could you please provide some data on how well the SQT are modelled compared to measurements? If no measurements are available from 2019 it would be good to show the mean modelled concentration compared to the observations from some other year.

- Lines 269–276, regarding the isoprene evaluation – there are isoprene measurements from about 20 sites in Europe during 2019 in ebas; a comparison of your model results to these data should be included in order to determine if the problems of overestimated isoprene emissions in MEGAN really are severe across Europe, or if it is more of a local problem in the forests around Hyytiälä.

- Lines 286–287: The sentence about the $PM_{2.5}$ comparison gives no useful information regarding the contents of the section (Analysis and source apportionment of OA). A proper model evaluation of $PM_{2.5}$ results should be included elsewhere in the manuscript (or not at all) and include sites all over Europe (not only in Spain and Italy, which makes no sense at all). The Supplemental

information Figure S1 and Table S2 both lack results from the $C_5H_8$-emissions-off simulation; this scenario should also be included, in case a more extended $PM_{2.5}$ model evaluation is included.

- Lines 290–293: "Extremely low OA concentrations are missed by the model, and there is a tendency of zeroing out such concentrations throughout the entire simulations (Figure 9 and Figure 10). The latest might suggest uncertainties in the background OA fields used in the model and/or in the concentrations injected at the very boundaries of the coarser domain (i.e., long-range transport)." These two sentences are unclear. As far as I can see from Fig. 9 and 10, the model produces lower OA concentrations than the measurements? What do you mean by "zeroing out such concentrations"? Regarding the suggested "uncertainties in the background OA fields" – how do you set the background OA concentrations in the model?

- Figure 13. It would be interesting to see the same type of plots for the large-scale model domain. Do the isoprene emissions have similar effects also in continental and southern Europe?

- Figure 15 and lines 361–362:  You suggest that the high night-time concentration of NOx in the model could be due to a "too shallow planetary boundary layer in the model". Is this a general problem in WRF-Chimere or a local problem in the region around Hyytiälä? Please provide evaluation against NOx measurements at other background stations in Europe, to make this manuscript of more general interest as a model evaluation study.

- Considering the overestimated NOx concentrations in the model (by a factor of two according to Table 4) – how does this influence the SOA-production? Will this lead to an overestimation of the "high-NOx"-path for SOA formation and underestimation of the "low-NOx"-path?

**Technical corrections**

There are a large number of minor language errors in the text and the manuscript would benefit from thorough language editing. I will not go through all language mistakes (I think that is the job of the author-team and possibly the language editing of the journal). I only list some of the mistakes I spotted here. In case the manuscript is revised, please make sure to have the language checked and corrected before resubmission.

- Line 29: "We attributed the latest" – what do you mean by that? Please reformulate.
- Line 72: "of the air mass" → "of the organic aerosol"?
- Line 88: "for the latest" – what do you mean? Please reformulate.
- Line 121, regarding the CAMS operational ensemble: I guess it is not WRF-Chimere that is part of the CAMS ensemble but rather the Chimere CTM using meteorology from the IFS model?
- Line 127 (and at many places in the manuscript): Your definition of the "astronomical summer" is not correct – please remove the word "astronomical" from the period description (it should be removed everywhere in the manuscript) – the astronomical summer of 2019 was 21 June – 23 September.
- Line 140. Oxidization → Oxidation
- Figure 2 caption, lines 763–765; the text "The text font size represents the tendency of both particles and gas-phase organic material (OM) to transition in the one or the other phase (i.e. larger font size indicated a better attitude towards that phase, and vice versa)" is awkward and needs rephrasing; you probably mean something like "stronger affinity" for the phases rather than "better attitude".

- Captions of Figures 5,6,7,11 and 12: change "the astronomical summer of 2019" to the actual time period included (15 Jun–30 Aug, 2019)

- Line 289: "flat diurnal of OA" → "flat diurnal variation of OA"
- Line 319: DMSP → DMPS
- Table 4 is not referred to anywhere in the text
- Table 4 and Table 5. SEMAR II → SMEAR II
- Lines 381 and 427: could → cloud
- Line 404: respond → response

---

## Referee Comment (RC2)

**General Comments**

This is a review of "On the formation of biogenic secondary organic aerosol in chemical transport models: an evaluation of the WRF-CHIMERE (v2020r2) model with a focus over the Finnish boreal forest" by Ciarelli et al., submitted to GMD. This paper investigates predictions from WRF-CHIMERE versus measurements at a boreal forest site (SMEAR II in Hyytiala Finland). The predictions compare well with most meteorological data (temperature, wind speed, RH, and wind direction) and gas phase species (monoterpenes, $O_3$, $NO_x$), but struggles with isoprene concentrations and precipitation events. The authors focus on the model's ability to predict biogenic secondary organic aerosol (BSOA) formation. To this end, the authors run a series of sensitivity simulations, altering the OH reaction rates and the isoprene emission rates. This manuscript could be improved by more clearly detailing which simulation is being discussed at any given time, and the goal and conclusions of running these different simulations (see below specific comment about this). This paper should be published after the below specific comments are addressed, and should be of interest to readers of GMD.

**Technical comments**

Section 2.2: can you add an explanation of *why* these different simulations are performed? I'm confused on the role of these simulations and what is discussed where in the following sections. The next time these simulations are mentioned as defined here is not until section 4.3/ line 303.

Line 234: can you quantify "a slight underestimation" in the text (from Table 3)? Can you speculate why this is occurring? It looks like the model predicts a lower nighttime temperature on almost every night except a handful, and actually does best during the heat wave, while capturing the daytime highs?

Line 236: it looks like the model misses all or almost all of the rain events, even the relatively large one during the heat wave, do you know why? Are they short-lived, or low total volume (i.e. do they have to last a specific amount of time or have a minimum volume to be captured)?

Line 242: an r value is provided for the wind speed, is it possible to also provide this for wind direction (on line 240 probably, or table 3)?

Line 250: what is causing the relatively high isoprene emissions in the "localized" area?

Figure 7: recommend making the percentages larger and bold, the text is small relative to the size of the wedges and hard to read.

Line 270: can you add a statistic to the text to quantify how much isoprene is "largely overestimated"? Either one of the values from Table 4 or something like number of days overestimated, average % overestimation, etc?

Figure 10: I don't think this figure adds much, suggest removing/moving to SI or combining with Figure 9

Line 296: this is the first time ASOA is mentioned in the body of the manuscript (not just the introduction), so suggest defining it again here

Figure 11: why is there a hot spot in POA over Turku but not ASOA?

Lines 294-305 & Figure 11: the discussion of ASOA feels misplaced since the discussion is focused on BVOC and BSOA up until here. Suggest adding some details to the methods section, or removing the discussion of ASOA.

Figure 12: similar to figure 7, suggest making text on the wedges larger

Section 4.4: I think this section would follow more logically if it was before current section 4.3?

Figure 14: I think this figure also might be removed or put in the SI

Line 360-361: can you quantify the diurnal O3 agreement and overestimation of NOx in the text (from table S4)?

Line 387: I assume "a.s.l." means "above sea level"? Suggest defining, and I'm not familiar enough to know if it's typically capitalized?

**Grammatical comments**

The manuscript is well written, although several minor grammatical errors exist throughout. While they do not impede the reader's understanding, the entire manuscript should be checked over before publication. Specific instances listed below, although please note I didn't not write them all down.

Line 23: "heat waves episodes"—waves should be singular

Line 73: "ration" should be "ratio"

Line 88&99: "where" should be "were"

Line 103 &140: "oxidization" should be "oxidation"

Line 194: "measurers" should be "measures"

Line 256: "there" should be "they"

Line 253: "measurement" should be plural

Line 257: "relatively" should be "relative"

Line 261: "differently" should be "different"

Line 262: "document" should be "documented"

Line 279: "instrumentation" should be "instrument" or could be removed entirely\

Line 295: "it is noticed" should be "is noticed"

Line 296: should "San Petersburg" be "Saint Petersburg"?

Line 314: "identify" should be "identified"

Line 320: "underestimate in the accumulation" should be "underestimate the accumulation"

Line 345: "increased" should be "increase"

Line 348: "over few regions" should be "over a few regions"

Line 349: "in the order" should be "on the order"

Line 350: "reacts" should be "react"

Line 354: "to have also important effect" should be "to also have important effects"

Line 376: "detailed" should be "details"

Line 379: "simulated period" should be "simulation period"

Line 379: "since, the latest, yields the" is worded awkwardly and parenthesis are misplaced. Maybe something like "since it yields the…"?

Line 386: "respect to" should be "with respect to"

Line 398: "slight" should be "slightly"

Line 392: I think "statistically-significant" should be "statistically significantly"?

---

## Author Comment (AC2)

**Replies to the Anonymous Referee 1**

We thank the referee for the valuable comments which helped us to improve the manuscript. Please find below our responses (in black) after the referee comments (in blue). Changes in the revised manuscript are written in *italics.*

This manuscript presents an evaluation and sensitivity analysis of the WRF-CHIMERE model, with a focus on biogenic secondary organic aerosol and the Hyytiälä monitoring site in Finland. While the topic is within the scope of GMD, the current version of the manuscript is not suitable for publication in the journal. The evaluation of the model is too limited to be of general interest, as it only covers a single site for two and a half months, which is not sufficient for a regional CTM model evaluation paper in GMD.

The paper does not present any substantial novel concepts, ideas, tools, or data, and does not represent a significant advance in modeling science. The short simulation period and apparently poor emission data for isoprene make it difficult to draw firm conclusions from the study, and there are few interpretations or conclusions presented in the manuscript.

If the model evaluation were extended to include more sites across Europe, it could be of sufficient interest to warrant publication in GMD. Organic carbon (OC) measurement data from 2019 are available from ebas.nilu.no for about 30 different regional sites in Europe, which would be a valuable addition to the evaluation. Given the poor agreement with observations for isoprene at Hyytiälä, it would be interesting to include isoprene measurements from other European sites (data from almost 20 sites are available for 2019 in ebas – including data from Pallas in Finland). It would also be useful to investigate the effect of isoprene emissions on ozone across Europe, using observations from the many regional background sites available in ebas. For some unclear reason, the comparison with observed PM2.5 measurements was restricted to sites in Spain and Italy (and this comparison is not discussed in detail in the manuscript). It would be interesting to evaluate the model's performance for PM2.5 across all of Europe (including Hyytiälä) and discuss these results in the manuscript.

In its current form, I recommend that the manuscript be rejected for publication in GMD. However, if the authors are willing to substantially extend their comparison with observations across Europe, their work may be reconsidered for publication.

We thank the referee for the additional remarks and suggestions on our manuscript. We agree with her/his suggestions, and we have now extended the model evaluation analysis to cover the whole European domain.

As the referee suggested, the manuscript has been substantially revised by using observational data across Europe using two specific databases, i.e., EBAS and the Air Quality e-Reporting database (i.e., Airbase). We believe that these suggestions greatly helped to corroborate the results currently presented for the Finnish Boreal Forest, which was initially selected in the study because of its high representativeness of biogenic aerosols formation processes.

Specifically, the revised manuscript now includes:

1) The evaluation of organic carbon (OC) and isoprene ($C_5H_8$) modeled fields over the European domain using observational data from the EBAS database (we provide specific discussion and details in the single replies below).

2) The evaluation of ozone ($O_3$), and nitrogen oxides ($NO_x$) modeled fields over the European domain using measurements data from the Air Quality e-Reporting database (i.e., Airbase) (we provide specific discussion and the details in the single replies below).

3) Additionally, we have extended the discussion of the results of the model sensitivity tests also to the European domain. The effect of inhibiting isoprene emissions on $O_3$, alpha-pinene ($C_{10}H_{16}$) and

biogenic secondary organic aerosol (BSOA) is now discussed across Europe in the main revised manuscript.

In the revised manuscript, we also further elucidated the reasons and motivation for the need of such model simulations analysis (as also asked by referee Nr. 2). Our evaluation study is built upon the increasingly comprehensive data sets available at supersite measurements stations like, for example, the Station for Measuring Ecosystem–Atmosphere Relations (SMEAR-II) located in the Finnish Boreal Forest, which provides a platform to evaluate model results to a great level of details thanks to parallel state-of-the-art measurements of a vast array of atmospheric compounds. As model simulations are growing in complexity, we believe that model evaluation studies are vital to support the modeling communities in reducing the sources of uncertainties in current biogenic secondary organic aerosols schemes and in the development of new numerical approaches, hopefully resulting in a better predictions of future climate scenarios.

The additional datasets used in the analysis is now described at page 8, line 218 of the revised manuscript ab below:

*Additional measurements of OC and isoprene air concentrations were taken from the EBAS European database (https://ebas.nilu.no/) (Table S1 and Table S2). $NO_x$ and $O_3$ measurements were retrieved for rural stations as available from the Air Quality e-Reporting (AQ e-Reporting) database (https://www.eea.europa.eu/en). Specifically, 271 stations were retrieved for $NO_x$ and 350 stations for $O_3$. Observations at these sites were compared against model data from the coarse grid (at 30 km). The statistical metrics used for the meteorological and chemical performance evaluation are reported in Table 2.*

And the following paragraph has been added in the introduction section at page 4, line 114 of the revisited manuscript:

*To provide a more comprehensive analysis of the simulations, model's results are additionally evaluated against observational data from two European databases, i.e., EBAS and the Air Quality e-Reporting (AQ e-Reporting) database.*

Specific comments

Section 2.2 only describes BSOA – how did you treat SOA from anthropogenic VOC? Did you include a VBS-treatment with (or without) aging also for ASOA?

We do include a VBS treatment of ASOA, and ASOA is allocated in the same range of volatilities as for BSOA but in different sets to uniquely separate the contribution of anthropogenic and biogenic compounds to secondary organic aerosol formation. Aging of ASOA is considered in our application with a reaction rate of $1 \times 10^{-11}$ molecule$^{-1}$ cm$^3$ s$^{-1}$ (Murphy and Pandis, 2009). This value is not altered when performing all the sensitivity tests. We added this additional information at page 6, line 176 of the revised manuscript as below:

*Formation of ASOA is included by using the same range of volatilities as for BSOA. Aging of ASOA is accounted for in our application with a reaction rate of $1 \times 10^{-11}$ molecule$^{-1}$ cm$^3$ s$^{-1}$ (Murphy and Pandis, 2009; Zhang et al., 2013). This value is not altered across all the sensitivity tests.*

You base your VBS scheme on Hodzic and Jimenez (2011) – their scheme only included SOA from OH-reactions. Did you include SOA formation from oxidation by ozone and/or NO3 radicals? Please provide details about how the BVOC + O3 and BVOC + NO3 are treated in the model.

We thank the reviewer for this comment. We would like to clarify that while the VBS included in CHIMERE was first implemented by Hodzic and Jimenez (2011), it is based on the works of Donahue et al., 2006; Lane et al., 2008a; Murphy and Pandis, 2009; Robinson et al., 2007, and the current version used in this study follows the implementation presented in details in the work of Zhang et al., 2013. Specifically, the formation

of biogenic SOA from $O_3$ and $NO_3$ following the same approach as in Murphy and Pandis, 2009) (Menut et al., 2021; Zhang et al., 2015, 2013). We added this additional information at page 6, line 156 of the revised manuscript as below:

*Additional formation of BSOA from $O_3$ and $NO_3$ is taking into account following the same approach as in Murphy and Pandis, 2009 (Menut et al., 2021; Zhang et al., 2015, 2013).*

Additionally, we re-phrased the sentence at page 5, line 141 as below to make it clearer that the VBS version used here is based on the work of Zhang et al., 2013.

*The VBS scheme was first implemented in the CHIMERE model for the Mexico City metropolitan area during the MILAGRO field experiment (Hodzic and Jimenez, 2011); however, the version included in the model is the one developed and applied over Europe for the Metropolitan area of Paris (Zhang et al., 2013).*

How did you treat SOA from SQT?

The model employs a rather simplified approach for the formation of BSOA from sesquiterpenes. BSOA from sesquiterpenes is considered only for reactions against OH radical, and oxidation products distributed in the same volatility bins as used for the rest of the BSOA precursors with not differentiation between low-NOx and high-NOx conditions. The reaction rate of sesquiterpenes against OH is set to 2.9 x $10^{-10}$ molecule$^{-1}$ cm$^3$ s$^{-1}$ and mass yields are taken from Tsimpidi et al., 2010. We added this additional information at page 5, line 147 of the revised manuscript as below:

*The model employs a simplified treatment for the formation of BSOA from sesquiterpenes. BSOA from sesquiterpenes is considered only for the reaction against the OH radical, and oxidation products distributed in the same volatility bins used for the rest of the BSOA precursors, and with not differentiation between low-NOx and high-NOx conditions. The reaction rate of sesquiterpenes (i.e., humulene) against OH is set to 2.9 x $10^{-10}$ molecule$^{-1}$ cm$^3$ s$^{-1}$ with mass yields from Tsimpidi et al., 2010.*

Hodzic and Jimenez (2011) only had a single monoterpene species (TERP) – you split the MT into four different species; please provide details of the differences in SOA yields (and reactivities) for the different MTs.

In our approach different MTs have identical SOA yields, but specific reactivities taken from Bessagnet et al., 2008. As mentioned above, the VBS used in this study has significant differences compared to Hodzic and Jimenez (2011). We added this additional information at page 5, line 146 of the revised manuscript as below:

*All monoterpene species have identical SOA yields, but specific reactivities based on Bessagnet et al., 2008.*

Hodzic and Jimenez (2011) also included SOA production from biomass burning, POA ageing, acidenhanced BSOA production and anthropogenic pollution-enhanced SOA production. Did you include all (or any) of these SOA formation reactions?

The CHIMERE model account for SOA production from biomass burning sources as well as aging of SVOC from primary organic aerosol (POA). SVOCs arising from the evaporation of POA are allowed to age with a reaction constant of 4 x $10^{-11}$ molecule$^{-1}$ cm$^3$ s$^{-1}$ (Robinson et al., 2007). We do not include specific acid enhanced BSOA production and anthropogenic pollution-enhanced SOA production. We added this additional information at page 6, line 178 of the revised manuscript as below:

*SVOCs arising from the evaporation of POA upon dilution are allowed to age with a reaction constant of 4 x $10^{-11}$ molecule$^{-1}$ cm$^3$ s$^{-1}$ (Robinson et al., 2007) and no acid enhanced BSOA production and anthropogenic pollution-enhanced SOA production is accounted for.*

Lines 146–147: You use ΔHvap of 36 kJ mol-1 – Hodzic & Jimenez 2011 used 88 kJ mol-1 – please explain why you chose the lower value.

We thank the reviewer for this comment. We would like to clarify that while the VBS included in CHIMERE was first implemented by Hodzic and Jimenez (2011), it is based on the works of Donahue et al., 2006; Lane et al., 2008a; Murphy and Pandis, 2009; Robinson et al., 2007, and the current version used in this study follows the implementation presented in details in the work of Zhang et al., 2013. Specifically, the enthalpy of evaporation was taken from the work of Murphy and Pandis, 2009 (30 kJ mol$^{-1}$), and kept identical for the application presented here. This values was selected to account for various temperature effects on SOA yields, and should be therefore considered as an "effective" enthalpy of evaporation, as also used in other modeling applications at European scale (Bergström et al., 2012; Ciarelli et al., 2016). Additionally, the value of 36 kJ mol$^{-1}$ is a typo in our manuscript, and it should read 30 kJ mol$^{-1}$ (Murphy and Pandis, 2009). We corrected the typo throughout the text. We revised the sentence at page 6, line 155 of the revised manuscript as below:

*The effective enthalpy of evaporation ($\Delta H_{vap}$) of each BSOA volatility class is unique and set to 30 kJ mol$^{-1}$.*

Did you only include the organic mass (including or excluding particulate water; or the full particle mass) when calculating the gas-particle partitioning of the SVOCs?

In the calculation of the gas-particle partitioning of SVOCs, the total particle mass is considered without including water.

How was the deposition of gas-phase SVOCs treated in the different model simulations?

For this applications, SVOCs wet depositions were kept identical among all the different simulations, i.e., the model does not account for the volatility dependence of the Henry's law water solubility coefficients. We added this additional information at page 6, line 180 of the revised manuscript as below.

*No volatility dependence of the Henry's law water solubility coefficients is included.*

Line 165: How were the annual anthropogenic emissions from CAMS "hourly distributed" for the simulation period? Please provide some details (and/or reference) regarding the temporal distribution of the emissions.

Temporal profiles are based on the EMEP MSC-W model temporal profiles (Simpson et al., 2012). The emissions are first distributed using a monthly profile, then they are distributed over the 24-hour day using a "day-type" profile (for specific countries and emission sectors). For each day of the week a specific profile is applied to consider weekday/weekend variations. We added this additional information in the paragraph at page 7, line 183 of the revised manuscript as below:

*Annual anthropogenic emissions of black carbon (BC), organic carbon (OC), carbon monoxide (CO), ammonia (NH$_3$), non-methane volatile organic compounds (NMVOCs), nitrogen oxides (NO$_x$) and sulphur dioxide (SO$_2$) were retrieved from CAMS for the whole year 2019 at 0.1 x 0.1-degree resolution and hourly distributed over the investigated periods (summer of 2019) with temporal profiles based on the EMEP MSC-W model (Simpson et al., 2012).*

Did you include any emissions from biomass burning (wildfires) in the simulations?

No emissions of wildfires were included in the simulation. We added this information at page 7, line 192 of the revised manuscript as below:

*No emissions from wildfires were included in the simulations.*

Lines 179–181: Were the initial and boundary concentrations of aerosols and gases taken from LMDz-INCA3 simulations for 2019 or for some other year? Were the boundary concentrations constant or varying in time?

Boundary conditions are taken from climatological global runs. Specifically, a monthly average of several years is created on a global level, which is then used as boundary conditions for the coarse domain. It should be considered that this study includes a nested simulation. For the nested domain, the coarse domain provides boundary/initial conditions. We added this additional information at page 7, line 199 of the revised manuscript as below:

*Initial and boundary conditions of aerosols and gas-phase constituents were retrieved from the climatological simulations of LMDz-INCA3 (Hauglustaine et al., 2014), where a monthly average of several years is created on a global level and used as boundary conditions for the coarse domain, and the Goddard Chemistry Aerosol Radiation and Transport (GOCART) model (Chin et al., 2002).*

Lines 198–204: The selection of PM2.5 data for the model evaluation is very odd for a study focussed on the "Finnish boreal forest". Also, there is no discussion of the results of the PM2.5 evaluation in the manuscript (only a figure and a table in the Supplement, with no accompanying text). As mentioned in my General comments, I think that the comparisons of modeled and measured PM2.5 should be made for all of Europe – and definitely include data from Hyytiälä. The Spanish and Italian sites included here are probably among the least interesting sites for the boreal Finnish forest so unless the PM2.5 evaluation is extended I think it should be removed completely.

We thank the reviewer for this remark. The selection of the southern European sites was an attempt to select stations likely to be exposed to higher level of oxidants, so to better probe into the effect the aging might induce on the aerosol mass. We agree with the reviewer that this approach was too much "sic et simpliciter" and the analysis has been removed but replaced with the direct evaluation of organic carbon (OC) data as available from the EBAS datasets. We believed that the suggestion of the reviewer strongly increased the evaluation analysis, which can now rely on both online high-resolution mass spectrometer data in the case of SMEAR-II and OC filters measurements as available from EBAS. The full discussion of this additional analysis is provided in the next comments below.

Lines 252–254: In what sense are your modeled BVOC emissions "generally in line with" the data presented by Hellén et al. (2018)? I do not think they are very much in line at all. For the summer 2016 they give (in their Table 1) mean total MT concentrations of 427 ppt and only 11 ppt isoprene and 13 ppt SQT; this seems very far from your ratios between the three BVOC types.

We apologies for the mistake. The sentence was to refer to the MT pool air concentrations, and not to the isoprene and the relative contribution of the single biogenic species. The sentence was removed since the direct comparison of MT and isoprene concentrations is directly provided in the timeseries of Figure 8 and Table 4 of the revised manuscript, and an additional comparison of sesquiterpene data is provided in the comment below as requested by the referee.

Figure 8 and Table 4. How does the statistics for the BVOCs change if you exclude the sawmill emission related time periods from the analysis? Does this improve the statistics for MT significantly?

There is a slightly improvement in the R values, i.e., up to 0.57 when excluding the sawmill emission related periods and the mean bias is reduced from -0.22 to -0.16 ppb.

Could you please provide some data on how well the SQT are modeled compared to measurements? If no measurements are available from 2019 it would be good to show the mean modeled concentration compared to the observations from some other year.

We agree with the reviewer, and we have compared the average modeled concentrations of sesquiterpenes at the SMEAR-II sites against the values reported in the study of Hellén et al., 2018 for the summer 2016, where the average of the detected sesquiterpenes concentrations was 13 ppt. Our modeled concentration

of SQT was found to be 15 ppt for the June – August 2019 period for the base case scenario. We added these values at page 10, line 287 of the revised manuscript as below:

*Modeled sesquiterpene concentration were found to be around 15 ppt on average for the investigated periods, which is comparable to the total detected sesquiterpenes average concentrations reported by Hellén et al., 2018 for the summer of 2016.*

Lines 269–276, regarding the isoprene evaluation – there are isoprene measurements from about 20 sites in Europe during 2019 in ebas; a comparison of your model results to these data should be included in order to determine if the problems of overestimated isoprene emissions in MEGAN really are severe across Europe, or if it is more of a local problem in the forests around Hyytiälä.

We agree with the referee, and we now extended our analysis also the European domain to better understand the overestimation of isoprene concentrations. For the investigated period, we were able to retrieve data for 10 additional stations from the EBAS data sets which include sufficient data and that are not located in urban/industrialized areas. Our results indicated that the overestimation is systematic across the majority of the European sites (Figure 1 below). Specifically, 70% of the analyzed stations reported an overestimation of isoprene air concentration. It is interesting to notice that also for the additional station located in Finland, i.e., Pallas (FI0096G) the model indicated a substantial overprediction of isoprene emissions (Figure 1 below), therefore indicating that the problem might be more accentuated for European boreal forests. This is also confirmed by a very recent global modeling study presented by Zhao et al., 2023 which used the GEOS-CHEM model over the northern high latitudes (Zhao et al., 2023).

[Figure]

Figure 1: Comparisons of modeled (red) and measured (blue) air concentrations of isoprene as available from the EBAS database. Measurement time resolution varies from is 1 hours to 4 days depending on the specific station. Units are in ppb vol.

We added the discussion of the additional results at page 11, line 291 of the revised manuscript as below, and included the additional comparison of isoprene concentrations in the Figure S1 of the supplementary information.

*An additional comparison with isoprene air concentration data as available from the EBAS database indicated that the overestimation is systematic across most of the European sites (Figure S1). Specifically, the model shows an overestimation of isoprene at 70% of the analyzed stations. It is interesting to notice that also for the additional station located in Finland, i.e., Pallas (FI0096G) the model indicated a substantial overprediction of isoprene emissions (Figure S1), therefore indicating that the problem might be more accentuated for European boreal forests. This is also confirmed by a very recent global modeling study presented by Zhao et al., 2023 where the GEOS-CHEM model was applied over the northern high latitudes (Zhao et al., 2023).*

Lines 286–287: The sentence about the PM2.5 comparison gives no useful information regarding the contents of the section (Analysis and source apportionment of OA). A proper model evaluation of PM2.5 results should be included elsewhere in the manuscript (or not at all) and include sites all over Europe (not only in Spain and Italy, which makes no sense at all). The Supplemental information Figure S1 and Table S2 both lack results from the C5H8-emissions-off simulation; this scenario should also be included, in case a more extended PM2.5 model evaluation is included.

We thank the reviewer for this remark.

As mentioned in the previous comments, the selection of the southern European sites was an attempt to select stations likely to be exposed to higher level of oxidant, so to better probe into the effects the aging processes might induce on the total aerosol mass when additional high-resolution parallel measurements of atmospheric constituent are not available. As the reviewer suggested, and we have now replaced this analysis with the direct evaluation of the modeled organic carbon (OC) mass against data as available from the EBAS datasets (from https://ebas.nilu.no/). In total, we were able to retrieve data from 15 additional sites at varying time resolution (ranging from 4 hours to 1 week). Since the model uses the organic aerosol (OA) mass concentrations in its own calculations, we apply the OA/OC ratio as in Bergström et al., 2012, which might however introduce additional uncertainties in the comparisons. Results are comparable with the analysis presented over the Finnish Boreal Forest, with the model showing a substantial increase in the OC mass and with larger overestimation for aging schemes that account for very aggressive aging processes (Figure 2, below). The mean bias varies from 0.63, -0.13 and -1.1 µg m$^{-3}$ for the Aging-On-Case1, Aging-On-Case2 and Aging-off case, respectively. We included the additional comparison of OC concentrations in Figure 10 of the revised main manuscript, and updated Table S1 and Table S3 of the revised supplementary material to include information on the location of the EBAS station and the statistic calculated according to the new analysis. We added the discussion of the additional evaluation in the paragraph at page 11, line 313 of the revised manuscript as below:

*We additionally compared model data against organic carbon (OC) measurements as available from 15 additional EBAS sites (Table S1) and at different time resolution (from 1 day to 1 week). Since the model uses the organic aerosol (OA) mass concentration in its own calculations, we applied the OA/OC ratio as in Bergström et al., 2012. Results indicated similar behaviors also for OC data (Figure 10) with the model showing a substantial increase in the OC mass, with larger overestimation for aging schemes that account for very aggressive aging processes. The mean bias varies from 0.63, -0.13 and -1.1 for the Aging-On-Case1, Aging-On-Case2 and Aging-off case, respectively (Table S3).*

[Figure]

**Figure 2: Model (y-axis) and measured (x-axis) air concentrations of OA for the Aging-On-Case-1 (left), b) Aging-On-Case-2 (center) and c) Aging-Off (right) BSOA schemes at the SMEAR-II station (upper pane, daily averages) and at available EBAS sites (bottom panel). Solid line indicates the 1:1 line. The dashed lines delimit 1:2 and 2:1 line. Units are in µg m⁻³. Data from the EBAS database has a time resolution varying from 4 hours to 1 week, depending on the specific site.**

Lines 290–293: "Extremely low OA concentrations are missed by the model, and there is a tendency of zeroing out such concentrations throughout the entire simulations (Figure 9 and Figure 10). The latest might suggest uncertainties in the background OA fields used in the model and/or in the concentrations injected at the very boundaries of the coarser domain (i.e., long-range transport)." These two sentences are unclear. As far as I can see from Fig. 9 and 10, the model produces lower OA concentrations than the measurements? What do you mean by "zeroing out such concentrations"? Regarding the suggested "uncertainties in the background OA fields" – how do you set the background OA concentrations in the model?

We agree with the referee that the sentence was not clear. The analysis of Figure 9 indicated that the model struggles in reproducing very low concentrations, which are often close to zero in the model output. In our simulation we do not set a specific background of OA within the domain, but we let the model calculate its own concentration based on current emissions and transportation processes. At the boundary of the nested domain, OA field are hourly injected based on the OA concentrations resolved in the parent grid (where boundary conditions are read from global simulations based on LMDz-INCA3 data). We have re-phrased the sentence at page 11, line 321 of the revised manuscript as below:

*Extremely low OA concentrations are missed by the model (Figure 9). The latest might suggest uncertainties in the background OA fields used in the model and/or in the concentrations injected at the very boundaries of the coarser domain (i.e., long-range transport).*

Figure 13. It would be interesting to see the same type of plots for the large-scale model domain. Do the isoprene emissions have similar effects also in continental and southern Europe?

We thank the referee for this comment which we believe largely enhanced the value of the paper. We have extended our analysis to additionally include the European domain. Figure 3 below shows that when isoprene is inhibited from the modeling system, O₃ concentrations are reduced mainly in continental and southern Europe. This is consistent with the enhanced photochemical activity in those areas, as well as of a large

availability of isoprene emission in the southern European regions (Curci et al., 2009). However, interesting, the increase in BSOA concentrations show an opposite pattern, and mainly interested the Scandinavian regions, where a large pool of alpha-pinene emissions is available. The decrease of alpha-pinene concentrations is evident all over the domain. Figure 13 (i.e., Figure 14 in the revised manuscript) has been updated by including the analysis across all Europe, and we have revised the paragraph at page 14, line 400 of the revised manuscript as below:

*Figure 14 reports the daytime relative changes in α-pinene, O₃ and BSOA concentrations between the two simulations performed with and without isoprene emissions across all Europe. Inhibiting isoprene emissions resulted in a non-negligible increase in the BSOA mass concentrations over larger areas of the northern part of the domain. In most of the areas, the BSOA mass increased by about 10 % with maximum increases at around 25 %. Conversely, α-pinene air concentrations were homogenously reduced all over the domain (Figure 14). The relative reductions (over land) were on the order of 10 to 20 %. As isoprene emissions are excluded from the modelling system, more α-pinene of biogenic origin can effectively react towards available radicals, i.e., ·OH radicals, and, owing to its higher mass yield compared to isoprene, effectively increase the production efficiency of BSOA. This process is likely favored by the large pool of α-pinene emissions available over the boreal forest regions (and by the lower temperatures compared to continental and southern Europe) which favors the transition of oxidized gases in the particle phase. Figure 14 also reports the relative changes in O₃ concentrations between the two simulations performed with and without isoprene emissions which were predicted to be very mild over the northern Europe and larger over continental and southern Europe because of enhanced photochemical activities and large availability of isoprene emissions in the southern regions of the domain.*

[Figure]

**Figure 3: Daytime average (08 - 20 LT) relative changes in $C_{10}H_{16}$ (alpha-pinene) air concentrations, Ozone and BSOA concentrations with and without isoprene emissions. The relative changes are calculated as ((C5H8-emissions-Off - Aging-On-Case-2) / Aging-On-Case-2) * 100.**

Figure 15 and lines 361–362: You suggest that the high night-time concentration of NOx in the model could be due to a "too shallow planetary boundary layer in the model". Is this a general problem in WRF-Chimere or a local problem in the region around Hyytiälä? Please provide evaluation against NOx measurements at other background stations in Europe, to make this manuscript of more general interest as a model evaluation study.

We agree with the reviewer, and we have now extended our analysis to the European domain using data from the Air Quality e-Reporting database (https://www.eea.europa.eu/en). In total, we were able to retrieve data for 271 rural stations for $NO_x$ and for 350 rural stations for $O_3$. The nighttime overestimation seems to be generalized (about 1 ppb, Figure 4 below). We have re-phrased the sentence at page 14, line 413 of the revised manuscript as below:

*As reported in the Figure 15, the model is capable to reproduce the diurnal variation and absolute values (ppb) of O₃ very well (mean bias of -0.1 ppb and 0.3 ppb for O₃ and NOₓ, respectively, Table 4), whereas NOx concentrations were overestimated during nighttime periods, a behavior that was also confirmed by an additional evaluation against NOₓ and O₃ measurements retrieved across whole Europe from the Air Quality e-Reporting database (https://www.eea.europa.eu/en) (Figure S3 and Table S4).*

[Figure]

[Figure]

**Figure 3: Diurnal variation of O₃ and NOₓ at available Air Quality e-Reporting rural sites (from 15 June until 30 August of 2019). Number of stations are 271 for NOₓ and 350 for O₃. The extent of the bars and the shaded areas denotes the one standard deviation (1σ). Measurements data are shown in in black and model data in red. Units are in ppb vol.**

Considering the overestimated NOx concentrations in the model (by a factor of two according to Table 4) – how does this influence the SOA-production? Will this lead to an overestimation of the "high-NOx"-path for SOA formation and underestimation of the "low-NOx"-path?

We thank the referee for this comment. Indeed, an uncertainty in modeled NOx concentrations will affect the formation of SOA, possibly reducing the SOA yields in case the high-NOx path will be favored. However, even though the analysis against the Air Quality e-Reporting database data seems to confirm a too strong dilution during daytime and a too shallow boundary layer height at night (Figure 4), decoupling the exact role of both meteorological, emissions, and chemical processes on the final modeled NOx concentrations remains challenging (especially at such low concentrations, below 1ppb at Hyytiälä), and an additionally analysis would be needed to specifically probe the role of these driving factors on NOₓ levels.

Technical corrections

There are a large number of minor language errors in the text and the manuscript would benefit from thorough language editing. I will not go through all language mistakes (I think that is the job of the author team and possibly the language editing of the journal). I only list some of the mistakes I spotted here. In case the manuscript is revised, please make sure to have the language checked and corrected before resubmission.

Line 29: "We attributed the latest" – what do you mean by that? Please reformulate.

The sentence was reformulated as below:

*Results indicated that the modeled BSOA concentrations generally increased compared to the base-case simulation with enabled isoprene emissions, possibly due to a shift in the reactions of monoterpenes compounds against available radicals, as further suggested by the reduction in α-pinene modeled air concentrations.*

Line 72: "of the air mass" → "of the organic aerosol"?

Corrected.

Line 88: "for the latest" – what do you mean? Please reformulate.

The sentence has been removed.

Line 121, regarding the CAMS operational ensemble: I guess it is not WRF-Chimere that is part of the CAMS ensemble but rather the Chimere CTM using meteorology from the IFS model?

That is correct and the sentence was reformulated as below at page 5, line 119 of the revised manuscript as below:

*The CHIMERE model has participated in numerous intercomparison exercises (Bessagnet et al., 2016; Ciarelli et al., 2019; Solazzo et al., 2017; Theobald et al., 2019) and it is an active member of the Copernicus Atmosphere Monitoring Service (CAMS) operational ensemble.*

Line 127 (and at many places in the manuscript): Your definition of the "astronomical summer" is not correct – please remove the word "astronomical" from the period description (it should be removed everywhere in the manuscript) – the astronomical summer of 2019 was 21 June – 23 September.

We agree with the referee, and we have corrected the occurrence everywhere in the manuscript.

Line 140. Oxidization → Oxidation

Corrected.

Figure 2 caption, lines 763–765; the text "The text font size represents the tendency of both particles and gas-phase organic material (OM) to transition in the one or the other phase (i.e. larger font size indicated a better attitude towards that phase, and vice versa)" is awkward and needs rephrasing; you probably mean something like "stronger affinity" for the phases rather than "better attitude".

Corrected.

Captions of Figures 5,6,7,11 and 12: change "the astronomical summer of 2019" to the actual time period included (15 Jun–30 Aug 2019).

Corrected.

Line 289: "flat diurnal of OA" → "flat diurnal variation of OA"

Corrected.

Line 319: DMSP → DMPS

Corrected.

Table 4 is not referred to anywhere in the text

We added the reference in the text at page 14 line 415 of the revised manuscript.

Table 4 and Table 5. SEMAR II → SMEAR II

Corrected.

Lines 381 and 427: could → cloud

Corrected.

Line 404: respond → response

Corrected.

**References**

[revised manuscript text omitted]

---

## Author Comment (AC3)

**Replies to the Anonymous Referee 2**

We thank the referee for the valuable comments which helped us to improve the manuscript. Please find below our responses (in black) after the referee comments (in blue). Changes in the revised manuscript are written in *italics.*

General Comments This is a review of "On the formation of biogenic secondary organic aerosol in chemical transport models: an evaluation of the WRF-CHIMERE (v2020r2) model with a focus over the Finnish boreal forest" by Ciarelli et al., submitted to GMD. This paper investigates predictions from WRF-CHIMERE versus measurements at a boreal forest site (SMEAR II in Hyytiala Finland). The predictions compare well with most meteorological data (temperature, wind speed, RH, and wind direction) and gas phase species (monoterpenes, O3, NOx), but struggles with isoprene concentrations and precipitation events. The authors focus on the model's ability to predict biogenic secondary organic aerosol (BSOA) formation. To this end, the authors run a series of sensitivity simulations, altering the OH reaction rates and the isoprene emission rates. This manuscript could be improved by more clearly detailing which simulation is being discussed at any given time, and the goal and conclusions of running these different simulations (see below specific comment about this). This paper should be published after the below specific comments are addressed, and should be of interest to readers of GMD.

We thank the referee for her/his comment on our manuscript. Our specific replies follow below.

Technical comments Section 2.2:

Section 2.2: can you add an explanation of *why* these different simulations are performed? I'm confused on the role of these simulations and what is discussed where in the following sections. The next time these simulations are mentioned as defined here is not until section 4.3/ line 303.

We thank the reviewer for this comment. Aging of biogenic aerosols have been evaluate in previous modeling application at European scale (Bergström et al., 2012; Cholakian et al., 2017; Zhang et al., 2013). However, very few of these studies have investigated the different effects of using varying biogenic aging scheme in an environment that is largely affected by biogenic emissions and by combining parallel measurements of biogenic precursors, meteorological parameters, and aerosol size distribution. We therefore performed an evaluation of the difference aging schemes, and underlying biogenic emission inventories, as currently available in literature, and by using the latest measurement data. Specifically, our evaluation study is built upon the increasingly comprehensive data sets available at supersite measurements stations like, for example, the Station for Measuring Ecosystem–Atmosphere Relations (SMEAR-II) located in the Finnish Boreal Forest, which provides a platform to evaluate model results to a great level of details thanks to parallel state-of-the-art measurements of a vast array of atmospheric compounds. As model simulations are growing in complexity, we believe that model evaluation studies are vital to support the modeling communities in reducing the sources of uncertainties in current biogenic secondary organic aerosols schemes and in the development of new numerical approaches, hopefully resulting in a better predictions of future climate scenarios.

We add the following paragraph at page 6, line 158 of the revised manuscript to better clarify the motivation for such analysis.

*Aging of biogenic aerosol have been tested in previous modeling application at European scale (Bergström et al., 2012; Cholakian et al., 2017; Zhang et al., 2013). However, very few of these studies have investigated the effects and impacts of using difference biogenic aging scheme in an environment that is largely affected by biogenic emissions and by combining parallel state-of-the-art measurements of a vast array of atmospheric*

*compounds. In this study we performed a comprehensive evaluation of the difference aging schemes as currently available from the literature.*

Line 234: can you quantify "a slight underestimation" in the text (from Table 3)? Can you speculate why this is occurring? It looks like the model predicts a lower nighttime temperature on almost every night except a handful, and actually does best during the heat wave, while capturing the daytime highs?

We thank the referee for this comment. Among the model resolution adopted here (about 10 km for the nested grid) which might influence the overall performance of the meteorological model, another important parameter that might affect model performance over large forest areas is the canopy effect. While the underlying emission model does account for the canopy effect, the current version of the CHIMERE model used in this study does not include any canopy effect. Recently, the canopy effect on vertical diffusion, wind speed, and shortwave radiation in the model was implemented and tested over the Landes pine forest in southwestern France (Cholakian et al., 2022) which we are planning to test also on the domain presented here.

We reported the value from Table 3 at page 9 line 253 of the revised manuscript as below:

*The model was able to reproduce such a temporal trend with a slight underestimation (-0.7°) occurring mainly during the nighttime periods (Figure 4).*

Line 236: it looks like the model misses all or almost all of the rain events, even the relatively large one during the heat wave, do you know why? Are they short-lived, or low total volume (i.e. do they have to last a specific amount of time or have a minimum volume to be captured)?

We thank the reviewer for her/his comments. We have additionally checked the rain intercomparison analysis. For the current analysis we were using modeled rain data as available from CHIMERE output in kg $m^{-2}$ $h^{-1}$, whereas the proper comparison be in mm $h^{-1}$ (which is written in the deposition files of CHIMERE). We have now reperformed the analysis, which revealed that the signal of the small rain event is actually captured by the model (Figure 4 below, scale has been adapted to facilitate the comprehension of the panel). However, the largest events are still missed (Figure 1, below), suggesting that the model might have difficulties to accurately reproduced short-lived events probably induced by local weather systems and the orographic processes specific of the site. We updated Figure 4 in the revised manuscript accordingly.

[Figure]

Line 242: an r value is provided for the wind speed, is it possible to also provide this for wind direction (on line 240 probably, or table 3)?

We agree with the reviewer, and we have now included the r value for the wind direction (r = 0.5) at page 10, line 258 of the revised manuscript as below:

*The analysis of the wind direction fields indicated that they were satisfactorily reproduced by the model, with the southern westerly (SW) sector being the most predominant wind direction during the summer period (r = 0.5).*

Line 250: what is causing the relatively high isoprene emissions in the "localized" area?

We believed this is likely caused by the underlying emissions factor associated with the vegetation data used with the MEGAN model, and more detailed analysis are planned to probe into those highly localized emissions in those areas (i.e., by using domain-specific land use).

Figure 7: recommend making the percentages larger and bold, the text is small relative to the size of the wedges and hard to read.

We modified Figure 7 as suggested by the referee in the revised manuscript.

Line 270: can you add a statistic to the text to quantify how much isoprene is "largely overestimated"? Either one of the values from Table 4 or something like number of days overestimated, average % overestimation, etc?

We agree with the reviewer, and we have quantified the overestimate in isoprene air concentration based on the Timeseries presented in Figure 8 of the manuscript. The ratio between the modeled and observed isoprene concentration varies from 4 to 8, with few isolated peaks exceeding a factor of 10. We have included this information at page 11, line 290 of the revised manuscript as below:

*The ratio between the modeled and observed isoprene air concentration varies from 4 to 8, with few isolated peaks exceeding a factor of 10.*

Figure 10: I don't think this figure adds much, suggest removing/moving to SI or combining with Figure 9

The referee is right. However, in the revised manuscript we have revised Figure 10 to also include the additional model evaluation for OC measurements as available from the EBAS datasets (as requested by referee nr. 1). For this reason, we prefer to keep the Figure 10.

Line 296: this is the first time ASOA is mentioned in the body of the manuscript (not just the introduction), so suggest defining it again here

We have re-defined the definition of ASOA (anthropogenic secondary organic aerosol) also at this occurrence in the revised manuscript.

Figure 11: why is there a hot spot in POA over Turku but not ASOA?

We thank the reviewer for this question. Turku is located on a coast side inside the domain. These sites are likely more challenging to resolve, at the current resolution, compared to other regions giving that the model cell grid needs to be resolved for both the water bodies and the physical terrain. Even though the underlying emissions inventories might lack several anthropogenic precursors which might not be fully resolve at the current resolution, the local meteorological condition can also highly influence the accumulation, production, and removal processes of secondary species. Indeed, higher ASOA concentration are visible over the area of Helsinki, and additional analysis would be needed to understand the differences in the formation of anthropogenic secondary organics aerosol (ASOA) at these two sites (at least from a modeling perspective).

Lines 294-305 & Figure 11: the discussion of ASOA feels misplaced since the discussion is focused on BVOC and BSOA up until here. Suggest adding some details to the methods section, or removing the discussion of ASOA.

We agree with the reviewer The treatment of ASOA in model is now discussed in detail in the revised manuscript in the methods section (as also asked by reviewer 1). The title of Section 2.2 has been changed as below:

*2.2 OA scheme*

Figure 12: similar to figure 7, suggest making text on the wedges larger

We modified Figure 12 as suggested by the referee in the revised manuscript.

Section 4.4: I think this section would follow more logically if it was before current section 4.3?

We agree with the reviewer, and we have now inverted the order of section 4.3 and 4.4 in the revised manuscript.

Figure 14: I think this figure also might be removed or put in the SI.

We agree with the reviewer, and we removed Figure 14 from the manuscript.

Line 360-361: can you quantify the diurnal O3 agreement and overestimation of NOx in the text (from table S4)?

We added the values from Table S4 at page 14, line 413 of the revised manuscript as below:

*As reported in the Figure 15, the model is capable to reproduce the diurnal variation and absolute values (ppb) of $O_3$ very well (mean bias of -0.1 ppb and 0.3 ppb for $O_3$ and $NO_x$, respectively, Table 4).*

Line 387: I assume "a.s.l." means "above sea level"? Suggest defining, and I'm not familiar enough to know if it's typically capitalized?

We agree with the reviewer and the have defined the acronym in the revisited manuscript.

Grammatical comments

The manuscript is well written, although several minor grammatical errors exist throughout. While they do not impede the reader's understanding, the entire manuscript should be checked over before publication. Specific instances listed below, although please note I didn't not write them all down.

Line 23: "heat waves episodes"—waves should be singular

Corrected.

Line 73: "ration" should be "ratio"

Corrected.

Line 88&99: "where" should be "were"

Corrected.

Line 103 &140: "oxidization" should be "oxidation"

Corrected.

Line 194: "measurers" should be "measures"

Corrected.

Line 256: "there" should be "they"

Corrected.

Line 253: "measurement" should be plural

Corrected.

Line 257: "relatively" should be "relative"

Corrected.

Line 261: "differently" should be "different"

Corrected.

Line 262: "document" should be "documented"

Corrected.

Line 279: "instrumentation" should be "instrument" or could be removed entirely\

We removed the work.

Line 295: "it is noticed" should be "is noticed

Corrected.

Line 296: should "San Petersburg" be "Saint Petersburg"?

Corrected.

Line 314: "identify" should be "identified"

Corrected.

Line 320: "underestimate in the accumulation" should be "underestimate the accumulation"

Corrected.

Line 345: "increased" should be "increase"

Corrected.

Line 348: "over few regions" should be "over a few regions"

Corrected.

Line 349: "in the order" should be "on the order"

Corrected.

Line 350: "reacts" should be "react

Corrected.

Line 354: "to have also important effect" should be "to also have important effects"

Corrected.

Line 376: "detailed" should be "details"

Corrected.

Line 379: "simulated period" should be "simulation period"

Corrected.

Line 379: "since, the latest, yields the" is worded awkwardly and parenthesis are misplaced. Maybe something like "since it yields the..."?

Corrected.

Line 386: "respect to" should be "with respect to"

Corrected.

Line 398: "slight" should be "slightly"

Corrected.

Line 392: I think "statistically-significant" should be "statistically significantly"?

Corrected.

**References**

Bergström, R., Denier van der Gon, H.A.C., Prévôt, A.S.H., Yttri, K.E., Simpson, D., 2012. Modelling of organic aerosols over Europe (2002–2007) using a volatility basis set (VBS) framework: application of different assumptions regarding the formation of secondary organic aerosol. Atmos. Chem. Phys. 12, 8499–8527. https://doi.org/10.5194/acp-12-8499-2012

Cholakian, A., Beekmann, M., Colette, A., Coll, I., Siour, G., Sciare, J., Marchand, N., Couvidat, F., Pey, J., Gros, V., Sauvage, S., Michoud, V., Sellegri, K., Colomb, A., Sartelet, K., Langley DeWitt, H., Elser, M., Prévot, A.S.H., Szidat, S., Dulac, F., 2017. Simulation of fine organic aerosols in the western Mediterranean area during the ChArMEx 2013 summer campaign. Atmos. Chem. Phys. Discuss. 2017, 1–45. https://doi.org/10.5194/acp-2017-697

Cholakian, A., Beekmann, M., Siour, G., Coll, I., Cirtog, M., Ormeno, E., Flaud, P.-M., Perraudin, E., Villenave, E., 2022. Simulation of organic aerosol, its precursors and related oxidants in the Landes pine forest in south-western France: Need to account for domain specific land-use and physical conditions (preprint). Aerosols/Atmospheric Modelling/Troposphere/Chemistry (chemical composition and reactions). https://doi.org/10.5194/acp-2022-697

Zhang, Q.J., Beekmann, M., Drewnick, F., Freutel, F., Schneider, J., Crippa, M., Prevot, A.S.H., Baltensperger, U., Poulain, L., Wiedensohler, A., Sciare, J., Gros, V., Borbon, A., Colomb, A., Michoud, V., Doussin, J.-F., Denier van der Gon, H.A.C., Haeffelin, M., Dupont, J.-C., Siour, G., Petetin, H., Bessagnet, B., Pandis, S.N., Hodzic, A., Sanchez, O., Honoré, C., Perrussel, O., 2013. Formation of organic aerosol in the Paris region during the MEGAPOLI summer campaign: evaluation of the volatility-basis-set approach within the CHIMERE model. Atmos. Chem. Phys. 13, 5767–5790. https://doi.org/10.5194/acp-13-5767-2013

---

## Author Response (AR2)

**Replies to the Anonymous Referee 1**

I am pleased to see that the manuscript has been substantially improved.

It is now of more general interest than the previous version (...). Some further comments/suggestions regarding the revised version, and the answers to my initial review are given below.

We thank the reviewer for the previous and additional comments on our manuscript. Please find below our additional responses (in black) after the referee comments (in blue). Changes in the revised manuscript are written in *italics.*

• "Additional formation of BSOA from O3 and NO3 is taking into account following the same approach as in Murphy and Pandis, 2009 (Menut et al., 2021; Zhang et al., 2015, 2013)."

This is still unclear to me. As far as I can see, none of the four references provide details about the SOA yields from the O3 and NO3-reactions of the VOCs. It is also a bit confusing to give four different references here – please stick to the one closest to your present implementation (I guess this would mean Zhang et al., 2013), in addition to the original Murphy and Pandis, 2009.

As far as I can see, the Zhang et al., 2013 study only used "low-NOx"-yields for all SOA-formation – is this true also for the present study? Or did you use different low-NOx and high-NOx yields depending on the NOx/VOC levels?

To make all these things clear, I suggest that you add a Table in the Supplement that gives the VBS-yields for the different SOA-precursors (isoprene, MT, SQT and the relevant AVOCs), and specify for which oxidants the yields are used – please also include information about low-NOx and high-NOx conditions in the table (if different yields are used for different conditions).

We agree with the referee, and we have now included a supplementary table to clarify and summarize all the details about the yields adopted in the VBS version of CHIMERE, i.e., Table 1 below (based on Zhang et al., 2013). Those yields applied to the whole SOA oxidants. For isoprene, only oxidation by OH are considered for BSOA formation. The differentiation of low-NOx and high-NOx is done as in Cholakian et al 2018 (which is based on the work of Zhang et al, 2013). Specifically, the yields of the two chemical regimes were calculated in the model with a tagged parameter (alpha) which calculates the ratio of the reaction rate of RO2 radicals with NO (high-NOx regime) with respect to the sum of reaction rates of the reactions with HO2 and RO2 (low-NOx regime). More information is available in Cholakian et al., 2018 and Carlton et al., 2009. Additionally, throughout the whole manuscript we now refer to the study of Zhang et al 2013 as main reference for the implementation (on top of the original Murphy and Pandis, 2009).

Table 1: Mass SOA yields for each SOA precursors used in the VBS version of CHIMERE. Yields are applied to all oxidants, i.e., $O_3$, $NO_3$, and OH. For isoprene, only oxidation by OH are considered for BSOA formation. The differentiation between low-NOx and high-NOx yield is based on the approached proposed by (Cholakian et al., 2018).

| SOA precursors | Mass yield of each bin | | | |
|---|---|---|---|---|
| $C*$ (µg m3) | 1 | 10 | 100 | 1000 |
| ALK4 | 0.0 | 0.075 | 0.0 | 0.0 |
| ALK5 | 0.0 | 0.300 | 0.0 | 0.0 |
| OLE1 | 0.0045 | 0.009 | 0.060 | 0.225 |
| OLE2 | 0.0225 | 0.435 | 0.129 | 0.375 |
| ARO1 | 0.075 | 0.225 | 0.375 | 0.525 |
| ARO2 | 0.075 | 0.300 | 0.375 | 0.525 |
| TERP | 0.1073 | 0.0918 | 0.3587 | 0.6075 |
| ISOP | 0.009 | 0.03 | 0.015 | 0.000 |
| SQT | 0.0750 | 0.1500 | 0.7500 | 0.9000 |

We added the following sentence at line 157 of page 6 of the revised manuscript as below:

*The complete list of the different SOA yields used in the model are reported in Table S1.*

• Q: How was the deposition of gas-phase SVOCs treated in the different model simulations? A: For this applications, SVOCs wet depositions were kept identical among all the different simulations, i.e., the model does not account for the volatility dependence of the Henry's law water solubility coefficients. We added this additional information at page 6, line 180 of the revised manuscript as below. "No volatility dependence of the Henry's law water solubility coefficients is included."

What about dry deposition? How did you treat dry deposition of gas-phase SVOCs?

The dry deposition of gases was treated with the Wesely model (Wesely, 1989). We added this information at line 182, page 6 of the revised manuscript as below:

*The dry deposition of gases was treated with the Wesely scheme* (Wesely, 1989)

• We additionally compared model data against organic carbon (OC) measurements as available from 15 additional EBAS sites (Table S1) and at different time resolution (from 1 day to 1 week). Since the model uses the organic aerosol (OA) mass concentration in its own calculations, we applied the OA/OC ratio as in Bergström et al., 2012.

This is unclear. Does the model track both OC and OM? If it only tracks OM – which OM/OC ratio(s) do you apply for the comparison to the OC-measurements? A single ratio for all types of OA? Or did you use different ratios for different types of SOA and POA?

We thank the reviewer for this comment. The model only tracks the OM mass. For this application we employed a single ratio for the OM/OC ratio, i.e., 1.7. We believe this is a fair approach for the comparison of model's results at rural sited during summer period. We revisited the sentence at line 316 page 11 as below:

*We applied the OM/OC ratio of 1.7 as representative for biogenic secondary organic aerosol (Bergström et al., 2012).*

• Extremely low OA concentrations are missed by the model (Figure 9). The latest might suggest uncertainties in the background OA fields used in the model and/or in the concentrations injected at the very boundaries of the coarser domain (i.e., long-range transport).

I think this is still confusingly formulated – do you mean that the model produces too low OA concentrations during periods with low measured OA? That is, that the model tends to underestimate OA for these periods? If so, perhaps you could write something like: The model tends to underestimate OA at Hyytiälä, especially during periods with low measured concentrations. Also, I think that the start of the second sentence "The latest" should be changed to "This".

We agree with the reviewer, and we reformulate the sentence at line 323 page 12 of the revisited manuscript as below as suggested by the reviewer:

*The model tends to underestimate OA at Hyytiälä, especially during periods with low measured concentrations (Figure 9). This might suggest uncertainties in the background OA fields used in the model and/or in the concentrations injected at the very boundaries of the coarser domain (i.e., long-range transport).*

Additional technical corrections:

Line 469: CAM -> CAMS

Corrected.

Figure 10 caption; Add information that the comparison for the ebas sites is for OC: "and OC at available EBAS sites".

Added.

Figure 12 caption: Add information that the results are for the SMEAR-II site

Added.

The Menut et al. (2021) reference should be updated to the final published version of the article (the manuscript version refers to the preprint)."

Corrected.

**Reference**

Bergström, R., Denier van der Gon, H.A.C., Prévôt, A.S.H., Yttri, K.E., Simpson, D., 2012. Modelling of organic aerosols over Europe (2002–2007) using a volatility basis set (VBS) framework: application of different assumptions regarding the formation of secondary organic aerosol. Atmos. Chem. Phys. 12, 8499–8527. https://doi.org/10.5194/acp-12-8499-2012

Carlton, A.G., Wiedinmyer, C., Kroll, J.H., 2009. A review of Secondary Organic Aerosol (SOA) formation from isoprene. Atmos. Chem. Phys. 9, 4987–5005. https://doi.org/10.5194/acp-9-4987-2009

Cholakian, A., Beekmann, M., Colette, A., Coll, I., Siour, G., Sciare, J., Marchand, N., Couvidat, F., Pey, J., Gros, V., Sauvage, S., Michoud, V., Sellegri, K., Colomb, A., Sartelet, K., Langley DeWitt, H., Elser, M., Prévot, A.S.H., Szidat, S., Dulac, F., 2018. Simulation of fine organic aerosols in the western Mediterranean area during the ChArMEx 2013 summer campaign. Atmos. Chem. Phys. 18, 7287–7312. https://doi.org/10.5194/acp-18-7287-2018

Wesely, M.L., 1989. Parameterization of surface resistances to gaseous dry deposition in regional-scale numerical models. Atmospheric Environment (1967) 23, 1293–1304. https://doi.org/10.1016/0004-6981(89)90153-4